# Advanced lipodystrophy reverses fatty liver in mice lacking adipocyte hormone-sensitive lipase

Laura Pajed[1,8], Ulrike Taschler [1,8], Anna Tilp[1], Peter Hofer[1], Petra Kotzbeck[2], Stephanie Kolleritsch[1], Franz P. W. Radner[1], Isabella Pototschnig[1], Carina Wagner[1], Margarita Schratter[1], Sandra Eder[1], Sabrina Huetter[1], Renate Schreiber [1], Guenter Haemmerle[1], Thomas O. Eichmann [1,3], Martina Schweiger [1], Gerald Hoefler[4,5], Erin E. Kershaw[6], Achim Lass [1,5] & Gabriele Schoiswohl [1,7 ✉]

Modulation of adipocyte lipolysis represents an attractive approach to treat metabolic diseases. Lipolysis mainly depends on two enzymes: adipose triglyceride lipase and hormone-sensitive lipase (HSL). Here, we investigated the short- and long-term impact of adipocyte HSL on energy homeostasis using adipocyte-specific HSL knockout (AHKO) mice. AHKO mice fed high-fat-diet (HFD) progressively developed lipodystrophy accompanied by excessive hepatic lipid accumulation. The increased hepatic triglyceride deposition was due to induced de novo lipogenesis driven by increased fatty acid release from adipose tissue during refeeding related to defective insulin signaling in adipose tissue. Remarkably, the fatty liver of HFD-fed AHKO mice reversed with advanced age. The reversal of fatty liver coincided with a pronounced lipodystrophic phenotype leading to blunted lipolytic activity in adipose tissue. Overall, we demonstrate that impaired adipocyte HSL-mediated lipolysis affects systemic energy homeostasis in AHKO mice, whereby with older age, these mice reverse their fatty liver despite advanced lipodystrophy.

[1] Institute of Molecular Biosciences, University of Graz, Graz, Austria. [2] Department of Internal Medicine, Division of Endocrinology and Diabetology, Medical University of Graz, Graz, Austria. [3] Center for Explorative Lipidomics, BioTechMed-Graz, Graz, Austria. [4] Diagnostic & Research Institute of Pathology, Medical University of Graz, Graz, Austria. [5] BioTechMed-Graz, Graz, Austria. [6] Division of Endocrinology and Metabolism, Department of Medicine, University of Pittsburgh, Pittsburgh, PA, USA. [7] Present address: Department of Pharmacology and Toxicology, University of Graz, Graz, Austria. [8] These authors contributed equally: Laura Pajed, Ulrike Taschler. ✉email: gabriele.schoiswohl@uni-graz.at

Adipose tissue plays an important role in the maintenance of whole-body energy homeostasis. Adipocytes evolved to store large amounts of fatty acids in form of energy dense triglycerides and to release them upon energy demand. Besides their function as systemic energy substrates, fatty acids serve as building blocks for cellular membranes, signaling molecules, and key mediators of various cellular processes. Deregulation of adipocyte lipid storage capacity, as observed under pathological conditions such as obesity or lipodystrophy, is associated with ectopic lipid accumulation and the development of "lipotoxic" metabolic disorders such as insulin resistance, glucose intolerance, and hepatic steatosis.

Lipolysis is a critical process controlling mobilization and storage of triglycerides. It is catalyzed by three lipases in consecutive steps[1]. Adipose triglyceride lipase (ATGL) initiates triglyceride hydrolysis releasing diglycerides and fatty acids. Hormone-sensitive lipase (HSL) preferentially degrades diglycerides to monoglycerides and fatty acids but is also able to hydrolyze triglycerides and monoglycerides. In the final step, monoglyceride lipase releases glycerol and the last fatty acid. Together, ATGL and HSL represent the main triglyceride hydrolases in adipose tissue[2]. Besides triglycerides, diglycerides, and monoglycerides, HSL also hydrolyzes cholesteryl esters and retinyl esters, representing the main hydrolase for diglycerides, cholesteryl esters, and retinyl esters in adipose tissue[3–5].

Impairment of adipocyte lipolysis via adipocyte-specific ATGL deletion improves systemic glucose homeostasis and protects from hepatic steatosis by reducing fatty acid delivery to non-adipose tissues, and subsequently, lipotoxicity[6–8]. Genetic and pharmacological inhibition of ATGL attenuates diet-induced obesity, mainly via hypophagia and reduced PPARg-mediated lipogenesis, lipid uptake, and synthesis[9,10]. Thus, reduced ATGL-mediated activity has a beneficial impact on systemic lipid and glucose homeostasis. In contrast, the impact of impaired HSL activity on the metabolic phenotype remained controversial. While HSL haplo-insufficiency or pharmacological inhibition of HSL improved glucose homeostasis and reduced hepatic triglyceride accumulation[11], adipose tissue-specific deletion of HSL resulted in glucose intolerance and fatty liver, despite reduced fatty acid release from adipose tissue into the circulation[12]. Similar to adipose tissue-specific HSL knockout (atHSL-KO) mice, humans with defective HSL have a higher risk for type 2 diabetes and hepatic steatosis[13], suggesting that metabolic dysregulations also in HSL-deficient humans are mainly driven by impaired adipocyte HSL activity. However, the underlying mechanisms remained elusive. Consistently, both humans and mice with HSL deficiency develop partial lipodystrophy associated with reduced PPARg signaling in adipose tissue[13–15]. Thus, the hepatic steatosis observed with HSL deficiency might be caused by insufficient lipid storage capacity in adipocytes lacking HSL. In support of this hypothesis, HSL-deficient humans have increased rather than decreased serum triglyceride levels[13]. In contrast, atHSL-KO mice do not have increased plasma triglycerides. Instead, lower fasting plasma fatty acid levels contribute to hepatic steatosis by reducing hepatic beta oxidation and very low-density lipoprotein (VLDL) synthesis[12]. Notably, previous studies in which adipocyte lipolysis was reduced through deficiency of ATGL or its coactivator alpha/beta hydrolase domain containing 5 (ABHD5; also known as comparative gene identification-58) also revealed reduced substrate availability for hepatic beta oxidation and VLDL synthesis[16]. However, in contrast to atHSL-KO mice, these mouse models are protected from hepatic steatosis[6,17]. Together, these data demonstrate a remarkable difference in the effect of specific lipolytic enzymes in adipose tissue on systemic and tissue-specific metabolic homeostasis and suggest additional mechanisms by which deficiency of HSL in adipocytes contributes to hepatic steatosis.

The aim of this study was to define the short- and long-term impact of HSL-mediated adipocyte lipolysis on energy and metabolic homeostasis in the setting of diet-induced obesity. We used adiponectin (Adipoq) driven Cre expressing transgenic mice harboring a floxed-Lipe gene encoding HSL, which lack enzyme activity in mature adipocytes[18]. We found that adipocyte-specific HSL knockout (AHKO) mice progressively develop a partial lipodystrophy despite reduced lipolytic activity. In early adulthood, AHKO mice had a fatty liver due to impaired insulin-mediated suppression of residual adipocyte lipolysis during refeeding. Remarkably, however, with advanced age, the fatty liver phenotype was reversed. Mechanistically, our data show that progressive lipodystrophy declines adipocyte lipolytic capacity in AHKO mice, thereby lowering the uncontrolled fatty acid release upon refeeding and improving glucose homeostasis. These data support a strong age-dependent effect of HSL-mediated adipocyte lipolysis on hepatic and systemic lipid metabolism.

## Results

**Adipocyte HSL deletion reduces lipolytic activity in white adipose tissue.** To investigate the role of HSL on adipose tissue function and systemic metabolism, we generated AHKO mice. AHKO mice had reduced Hsl mRNA and protein expression exclusively in white and brown adipose tissue (WAT: perigonadal adipose tissue (PGAT) and subcutaneous adipose tissue (SCAT); BAT), but not in non-adipose tissue such as liver (Fig. 1a). As expected, the loss of HSL in adipocytes caused reduced basal and β-adrenergic stimulated lipolysis both in vivo (Fig. 1b) and in ex vivo fat pad assays using PGAT explants (Fig. 1c). As a result of impaired lipolytic activity in WAT, AHKO mice showed lower plasma fatty acid and glycerol levels in the fasted state (Fig. 1d). Thus, adipocyte-specific HSL deletion reduces fasting-induced lipolysis of WAT.

**HFD-fed AHKO mice develop a lipodystrophic phenotype accompanied by increased immune cell infiltration and reduced expression of PPARg and its target genes regulating lipid metabolism.** It was recently shown that the loss of HSL in adipose tissue results in age-dependent partial lipodystrophy in chow-fed mice[12]. To investigate whether this phenotype was more pronounced when challenged with diet-induced obesity, we fed AHKO mice a HFD for 6 months. At an age of 24 weeks, AHKO mice showed lower body weight (Fig. 2a). This was accompanied by reduced gain of fat mass, starting at an age of 16 weeks (Fig. 2b), which was an earlier onset than previously observed in chow-fed mice[12]. Reduced fat mass was partially compensated by increased lean mass (Fig. 2c) resulting from higher liver, skeletal muscle, pancreas, and spleen weight (Fig. 2d). Fasted, but not ad libitum fed AHKO mice on HFD had lower levels of plasma fatty acids, glycerol, triglycerides, and ketone bodies compared to control mice fed HFD (Table 1). Despite changes in body weight and body composition, total energy expenditure, oxygen consumption, and locomotive activity were unchanged between the genotypes (Supplementary Fig. 1a–c). Likewise, food intake of AHKO mice on HFD was not different compared to control mice (Fig. 2e and Supplementary Fig. 1d), indicating that reduced caloric intake was not the cause of the observed lipodystrophic phenotype of AHKO mice. Furthermore, feces output and the corresponding fecal energy content as analyzed by calorimetric analysis were not different between AHKO and control mice (Supplementary Fig. 1e). These data indicate sufficient intestinal food absorption in AHKO mice.

Since fat deposition in HFD-fed AHKO mice declined with age, we compared the adipose tissue phenotype of young (10 weeks of age) and middle-aged (26 weeks of age) mice.

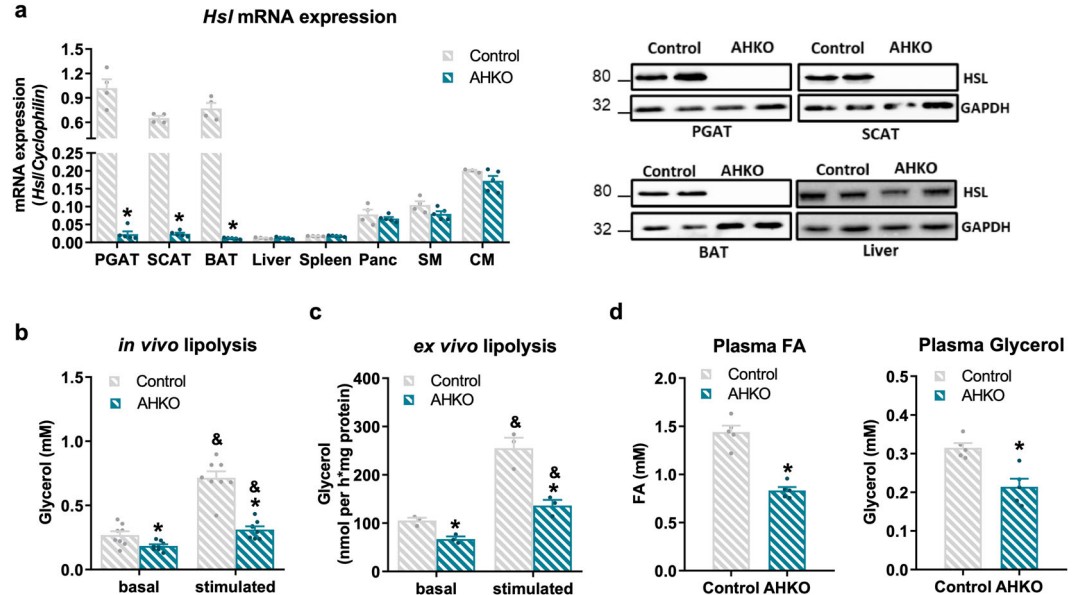

**Fig. 1 Effects of adipocyte-specific HSL deletion on adipocyte lipolysis. a** Left: *Hsl* mRNA expression relative to *Cyclophilin* reference gene by qPCR in adipose tissue (PGAT perigonadal, SCAT subcutaneous, BAT brown) and non-adipose tissue (Panc pancreas, SM skeletal muscle [quadriceps], CM cardiac muscle) with *Hsl* expression in control PGAT arbitrarily set to 1. Right: HSL protein expression in adipose tissue and liver (M, 33 weeks, chow, 12 h fasted, *n* = 4–5 animals/genotype). **b** In vivo lipolysis determined by plasma glycerol levels before (basal) and 15 min (stimulated) after ip administration of 1 mg kg$^{-1}$ body weight of the $\beta_3$-adrenergic receptor agonist CL-316,243 (M, 10 weeks, chow, 6 h fasted, *n* = 8 animals/genotype). **c** Ex vivo lipolysis determined by glycerol released from PGAT explants incubated without (basal) and with (stimulated) 10 μM isoproterenol for 1 h (M, 12 weeks, chow, ad libitum fed, *n* = 3 animals/genotype). **d** Plasma fatty acid (FA; left) and glycerol levels (right) (M, 26 weeks, chow, 12 h fasted, *n* = 5 animals/genotype). Data represent mean + SEM. Statistical significance was determined by Student's two-tailed *t* test. *P* < 0.05: * for effect of genotype; & for effect of treatment.

BAT weight was comparable in young HFD-fed AHKO and control mice, but higher in middle-aged HFD-fed AHKO mice (70%; Fig. 2f). Conversely, WAT weight of PGAT and SCAT was similar at an age of 10 weeks but was lower in middle-aged AHKO mice (PGAT: −90%; SCAT: −40%) compared to control mice (Fig. 2f). WAT mass conformed with plasma leptin concentrations, which were unchanged in young (5 ± 1 vs. 7 ± 2 ng/ml) but reduced in middle-aged AHKO mice (47 ± 9 vs. 18 ± 2 ng/ml; *P* = 0.006). Furthermore, WAT from AHKO mice exhibited increased mRNA expression of immune cell markers such as *F4/80* and *Cd11c* (Fig. 2g), which was most apparent in PGAT and SCAT of middle-aged AHKO mice. In accordance, perilipin 1 protein expression, a key marker for healthy adipocytes[19], was largely reduced in both PGAT and SCAT of young and middle-aged AHKO mice (Supplementary Fig. 1f).

In accordance with unchanged WAT weight of young AHKO mice, only minor differences were observed in the expression of genes involved in lipid metabolism in PGAT (Fig. 2h). Yet, prolonged adipocyte HSL deficiency, as in middle-aged AHKO mice, led to pronounced dysregulation of lipid metabolism (Fig. 2h), including downregulation of mRNA levels of *Pparg2* and genes involved in lipogenesis and/or adipogenesis (*Cebp1a*, *Srebp1c*), lipid uptake (*Cd36*, *Fabp4*), and lipid synthesis (*Pepck*, *Fasn*, *Dgat2*). Notably, changes were more pronounced in PGAT which also showed a more lipodystrophic phenotype compared to SCAT in middle-aged AHKO mice (Fig. 2f). To investigate whether the reduction in adipose tissue mass was associated with a switch to a browning/beiging phenotype, we further determined mRNA expression of genes involved in thermogenesis. While *Ucp1* and *Dio2* were or tended to be upregulated in PGAT and SCAT, the expression of *Pgc1a*, *Cidea*, and *Prdm16* was reduced in PGAT or SCAT (Supplementary Fig. 1g). Despite induction of *Ucp1* mRNA expression in both WAT depots, we were not able to

detect UCP1 protein in PGAT or SCAT, suggesting that increased thermogenesis had no or only a minor impact on the reduction of WAT mass. Thus, adipocyte-specific HSL deletion impairs age- and HFD-dependent adipose tissue expansion via impaired PPARg-regulated processes such as lipogenesis, lipid synthesis, and lipid uptake accompanied by a progressive inflammatory response.

**HSL deficiency in adipocytes impairs adipose tissue insulin signaling and insulin-mediated suppression of fatty acid release leading to fatty liver.** Hepatic triglyceride content was similar between young control and AHKO mice on HFD but increased in middle-aged AHKO mice (Fig. 3a). Furthermore, expression of marker genes for hepatic fibrosis (*Col1a1*, *Col1a2*, *Tgfb*) and the immune cell marker *Cd11c* was higher in liver of AHKO mice fed HFD (Fig. 3b). To investigate how HSL deficiency induces hepatic lipid accumulation in HFD-fed mice, we determined plasma fatty acid levels upon fasting and at different time-points after refeeding. In control mice, plasma fatty acid levels were high upon fasting and declined after 2, 4, and 8 h of refeeding (Fig. 3c). As expected, AHKO mice exhibited lower fasting plasma fatty acid levels than control mice, consistent with impaired lipolysis and release of fatty acids from adipose tissue. Upon refeeding, however, plasma fatty acid levels remained high in AHKO mice, and were even higher than in control mice. To evaluate whether the increased plasma fatty acid levels were due to impaired lipid uptake, we determined systemic and adipose tissue-specific lipid clearance following an olive oil gavage (Supplementary Fig. 2a, b). However, both, the systemic and tissue-specific lipid uptake were similar between the genotypes, suggesting that AHKO mice rather have impaired suppression of adipose tissue lipolysis upon refeeding than reduced lipid uptake. Correspondingly, PGAT and SCAT of AHKO mice on HFD

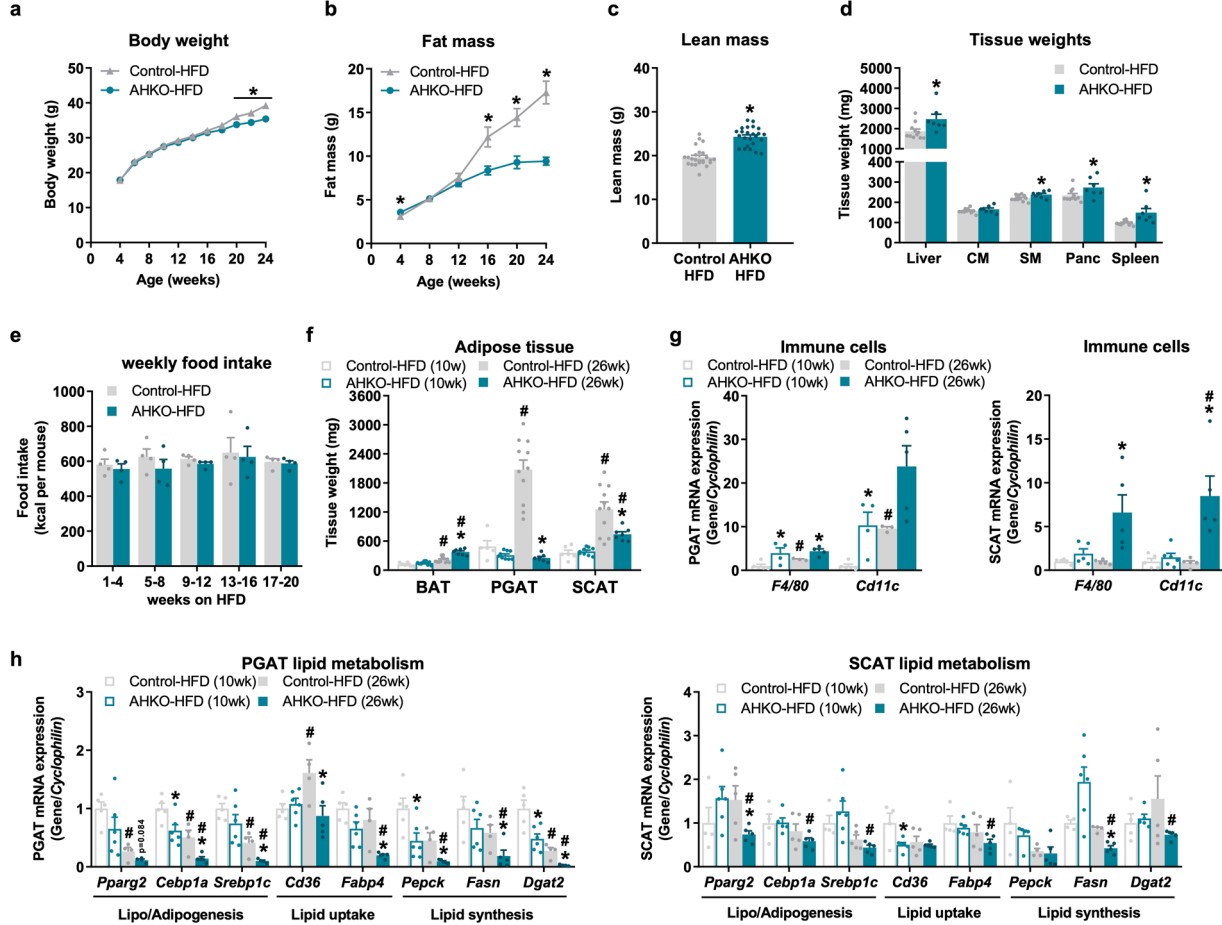

**Fig. 2 Effects of adipocyte-specific HSL deletion on systemic energy homeostasis and adipose tissue phenotype in HFD-fed mice. a** Longitudinal body weight and **b** fat mass in control and AHKO mice fed HFD (M, 4–24 weeks, ad libitum fed, $n = 9$–25 animals/genotype). **c** Lean mass (M, 24 weeks, HFD, ad libitum fed, $n = 24$ animals/genotype). **d** Tissue weights of non-adipose tissues (Liver, CM cardiac muscle, SM skeletal muscle [quadriceps], Panc pancreas, spleen) (M, 26 weeks, HFD, ad libitum fed, $n = 7$–11 animals/genotype). **e** Weekly food intake in HFD-fed control and AHKO mice (M, ad libitum fed, $n = 4$ animals/genotype). **f** Adipose tissue (BAT brown, PGAT perigonadal, SCAT subcutaneous) weight of 10- and 26-week-old mice (M, HFD, ad libitum fed, $n = 6$–11 animals/genotype). **g** PGAT (left) and SCAT (right) mRNA expression of immune cell markers and **h** genes involved in lipid metabolism relative to *Cyclophilin* reference gene by qPCR with the young (10 weeks) control animals arbitrarily set to 1 for each gene (M, 10 and 26 weeks, HFD, ad libitum fed, $n = 4$–6 animals/genotype). Data represent mean ± SEM. Statistical significance was determined by Student's two-tailed *t* test. *P* values compare effect of genotype; $P < 0.05$: * for effect of genotype; # for effect of age.

**Table 1 Plasma parameters of control and AHKO mice fed HFD.**

| Parameter | Control-HFD | AHKO-HFD | Control-HFD | AHKO-HFD |
|---|---|---|---|---|
| | Fed | | Fasted | |
| Fatty acids (mM) | 0.64 ± 0.06 | 0.70 ± 0.05 | 0.91 ± 0.15 | 0.54 ± 0.05* |
| Glycerol (mM) | 0.33 ± 0.04 | 0.31 ± 0.05 | 0.44 ± 0.07 | 0.18 ± 0.02* |
| Triglycerides (mM) | 0.95 ± 0.07 | 1.12 ± 0.21 | 0.76 ± 0.06 | 0.32 ± 0.02* |
| Ketone bodies (mM) | n.d. | n.d. | 0.97 ± 0.19 | 0.45 ± 0.08* |

Metabolites were determined in control and AHKO mice in the ad libitum fed or 12h-fasted state (M, 26 weeks, HFD, $n = 4$–10 animals/genotype). Data represent mean ± SEM. Statistical significance was determined by Student's two-tailed *t* test.
*n.d.* not determined.
*$P < 0.05$; for effect of genotype.

exhibited reduced insulin-mediated phosphorylation of AKT$^{P-Ser473}$ (Fig. 3d). Furthermore, primary adipocytes obtained from AHKO adipose tissue showed impaired anti-lipolytic response to insulin (Supplementary Fig. 2c), indicating that AHKO adipose tissue was insulin resistant. Consistently, plasma insulin levels of AHKO mice drastically increased upon refeeding (Fig. 3c).

Together, increased plasma fatty acid and insulin levels suggest that adipose tissue of AHKO mice is insulin resistant, resulting in incomplete suppression of residual adipocyte lipolysis. Notably, these elevated plasma fatty acid and insulin levels in AHKO mice upon refeeding were already observed in young AHKO mice (Supplementary Fig. 2d), indicating that insulin resistance and

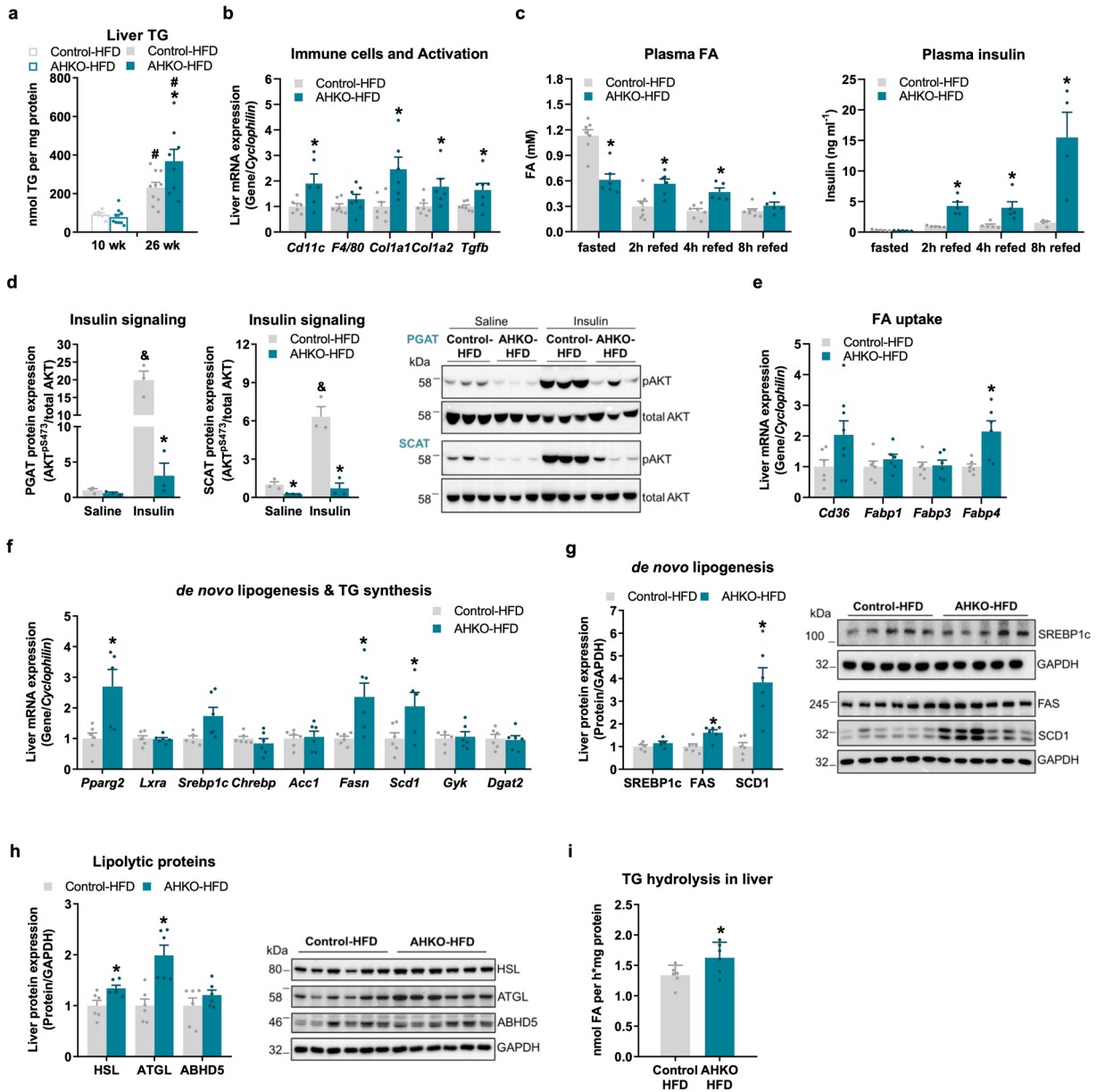

**Fig. 3 Effects of adipocyte-specific HSL deletion on hepatic lipid homeostasis. a** Hepatic triglyceride (TG) content (M, 10 and 26 weeks, HFD, ad libitum fed, $n = 6$–10 animals/genotype). **b** Liver mRNA expression of immune cell markers and genes involved in fibrosis relative to *Cyclophilin* reference gene by qPCR with control animals arbitrarily set to 1 for each gene (M, 26 weeks, HFD, ad libitum fed, $n = 5$–6 animals/genotype). **c** 12 h fasted and refed plasma fatty acid (FA; left) and insulin (right) levels (M, HFD, 17 weeks, $n = 4$–7 animals/genotype). **d** Insulin signaling in WAT. Mice were injected with saline or insulin at 0.75 IU kg$^{-1}$ body weight. Left: quantification of phosphorylation of AKT$^{pSer473}$/total AKT in PGAT and SCAT (M, 26 weeks, HFD, 12 h fasted, $n = 3$ animals/genotype). Right: representative immunoblots. **e** Hepatic mRNA expression of FA transporter in the liver (M, 22 weeks, HFD, ad libitum fed, $n = 6$–8 animals/genotype) and **f** genes involved in de novo lipogenesis and triglyceride (TG) synthesis relative to *Cyclophilin* reference gene by qPCR with control animals arbitrarily set to 1 for each gene (M, 26 weeks, HFD, ad libitum fed, $n = 5$–6 animals/genotype). **g** Left: liver expression of proteins involved in de novo lipogenesis (SREBP1c, FAS, SCD1). Right: representative immunoblots (M, 26 weeks, HFD, ad libitum fed, $n = 6$ animals/genotype). **h** Left: liver protein expression of lipolytic proteins (HSL, ATGL, ABHD5). Right: representative immunoblots (M, 26 weeks, HFD, ad libitum fed, $n = 6$ animals/genotype). **i** Triglyceride hydrolase activity in liver determined by [$^3$H]-labeled oleic acid release (M, 26 weeks, HFD, ad libitum fed, $n = 4$–5 animals/genotype). Data represent mean + SEM. Statistical significance was determined by Student's two-tailed $t$ test. $P < 0.05$: * for effect of genotype; # for effect of age; & for effect of treatment.

impaired suppression of lipolysis developed already at an early stage.

Since AHKO mice on HFD had higher plasma fatty acid levels upon refeeding, we investigated whether that provoked enhanced hepatic fatty acid uptake and consequently the development of a fatty liver. mRNA expression of fatty acid transporter *Fabp4* was

and *Cd36* tended to be higher in liver of AHKO mice (Fig. 3e). In line with the upregulation of these two PPARg target genes, the expression of *Pparg2* itself was 2.7-fold higher in liver of HFD-fed AHKO mice compared to control mice (Fig. 3f). Increased *Pparg2* mRNA expression was accompanied by higher expression of *Fasn* and *Scd1*, key target genes involved in hepatic de novo

lipogenesis, while expression of *Srebp1c* was moderately, but not significantly, increased (Fig. 3f). Consistently, FAS and SCD1 protein expression were increased, while SREBP1c protein expression was unchanged in AHKO liver (Fig. 3g). Hepatic expression of genes involved in triglyceride synthesis (*Gyk, Dgat2*, Fig. 3f), beta oxidation (*Ppara, Cpt1a, Lcad*; Supplementary Fig. 2e), and glucose metabolism (*Glut2, Gck, Pgc1a, Pepck, G6Pase*; Supplementary Fig. 2f) was comparable between genotypes. Notably, protein expression of HSL and ATGL was upregulated in liver (Fig. 3h), resulting in a moderate increase in triglyceride hydrolase activity (Fig. 3i). Thus, the fatty liver phenotype observed in HFD-fed AHKO mice is mainly orchestrated by the insulin-resistant adipose tissue of AHKO mice via increased flux of fatty acids from adipose tissue to the liver upon refeeding and associated with the induction of de novo lipogenesis.

**Blunted lipolytic activity in AHKO adipose tissue reverses liver phenotype in aged mice.** Since AHKO mice progressively developed a fatty liver, we investigated the long-term effect of adipocyte HSL deficiency on the liver phenotype. AHKO mice fed HFD up to 44 weeks of age (aged mice) exhibited similar energy expenditure compared to control mice (Supplementary Fig. 3a), but oxygen consumption rate was reduced in the light phase, while locomotive activity tended to be increased during the dark phase (Supplementary Fig. 3b, c). This increment might be caused by increased food intake during the dark phase (Supplementary Fig. 3d). However, despite increased food intake, AHKO mice had reduced body weight (Fig. 4a) and much lower fat mass (−70%; Fig. 4b) than control mice. Thus, lipodystrophy of aged AHKO mice was much more pronounced compared to middle-aged AHKO mice (Fig. 2b). This difference was primarily caused by a decline in SCAT weight (−80%), while PGAT weight (−90%) did not further decrease (Fig. 4c). In contrast, non-adipose tissue weights including skeletal muscle, pancreas, and spleen were increased, but against our expectations, liver weight was reduced in aged AHKO mice (Fig. 4d). Thus, triglyceride content did not further increase in the liver of aged AHKO mice opposed to aged control mice, which showed an age- and HFD-dependent accumulation of triglycerides in the liver leading to higher hepatic triglyceride levels in aged control mice compared to AHKO mice (Fig. 4e). In line with reduced hepatic lipid accumulation the mRNA expression of immune cell markers (*Cd11c, F4/80*) and genes involved in fibrosis (*Col1a1, Col1a2, Tgfb*) tended to be downregulated in aged AHKO liver (Fig. 4f). Gross and microscopic staining of the liver corroborated the age-dependent lipid accumulation and a corresponding increase in F4/80-positive cells in aged control mice compared to age-matched AHKO mice (Fig. 4g). Accordingly, plasma ALT levels tended to be higher in middle-aged AHKO mice (15.6 ± 2.0 vs. 36.6 ± 6.2 ng/ml; *P* = 0.074) but were comparable between aged AHKO and control mice (25.5 ± 3.3 vs. 24.5 ± 4.5 ng/ml), suggesting that liver of AHKO mice recovered with age.

Next, we determined plasma fatty acid levels in the fasted and refed state in aged AHKO and control mice. As expected, fasted plasma fatty acid levels of AHKO mice on HFD were lower compared to control mice (Fig. 4h), a phenotype that had also been observed in fasted young and middle-aged AHKO mice (Supplementary Fig. 2d and Fig. 3c). In contrast to middle-aged AHKO mice, however, aged AHKO mice did not show elevated plasma fatty acid levels upon refeeding compared to control mice, while plasma insulin levels were still elevated in refed AHKO mice (Fig. 4h). Since refed fatty acid levels were comparable between the genotypes, we hypothesized that insulin-mediated suppression of adipocyte lipolysis improved with age in AHKO

mice. However, insulin-mediated phosphorylation of AKT$^{pSer473}$ was still declined in WAT of aged AHKO mice (Fig. 4i and Supplementary Fig. 3e). Consistent with unchanged plasma fatty acid levels of AHKO mice between fasting and refeeding, protein expression of ATGL was significantly reduced in PGAT of aged AHKO mice compared to control mice or to middle-aged AHKO mice (Fig. 4j). In contrast, ABHD5 protein expression was downregulated in both middle-aged and aged AHKO mice. In line, triglyceride hydrolase activity was reduced in PGAT of middle-aged and aged AHKO mice, but the reduction of lipolytic activity was more pronounced in aged mice (−60% compared to middle-aged AHKO mice; Fig. 4k), indicating that aged AHKO WAT (PGAT) loses its lipolytic ability.

Consistent with unchanged plasma fatty acid levels of refed AHKO mice, mRNA expression of *Pparg2* as well as most investigated genes involved in de novo lipogenesis and triglyceride synthesis (*Lxra, Srebp1c, Chrebp, Acc1, Scd1, Dgat2*) was unchanged in liver of AHKO mice, with the exceptions of decreased PPARg targets *Fasn, Cd36*, and *Fabp4* as well as *Gyk* (Fig. 5a, b). Likewise, protein expression of FAS and SCD1 but also ACC was reduced in liver lysates of AHKO mice (Fig. 5c). The expression of genes involved in beta oxidation (*Ppara, Cpt1a, Lcad*, Supplementary Fig. 4a), protein expression of lipolytic enzymes (HSL, ATGL, ABHD5; Supplementary Fig. 4b), as well as hepatic triglyceride hydrolase activity (Supplementary Fig. 4c) were all unchanged in the liver of AHKO mice compared to control mice. Again, mRNA expression of genes involved in glucose metabolism (*Glut2, Gck, Pgc1a, Pepck, G6Pase*; Supplementary Fig. 4d) was not different in the liver of AHKO mice compared to control mice. In summary, these data demonstrate that fatty liver of middle-aged AHKO mice normalized at older age, presumably because the lipodystrophic phenotype had further progressed, which blunted lipolytic capacity, as evident by unchanged plasma fatty acid levels of fasted and refed AHKO mice. Interestingly, PPARg2 but not SREBP1c expression was elevated in middle-aged vs. aged AHKO liver, indicating that hepatic PPARg2 was the driver of increased de novo lipogenesis and the pathogenesis of fatty liver in these mice.

**Aged AHKO mice with advanced lipodystrophy show improved glucose clearance.** To investigate whether the loss of adipose tissue mass due to advanced lipodystrophy and the absence of the lipolytic fasting response translate into improved glucose utilization, we investigated glucose tolerance and insulin sensitivity of HFD-fed AHKO mice at different age. Blood glucose levels of middle-aged and aged AHKO mice, irrespective of the feeding status, were comparable to control mice (Fig. 6a). In contrast, insulin levels were higher in ad libitum fed but lower in fasted middle-aged and aged AHKO mice compared to control mice (Fig. 6b). Accordingly, insulin levels were reduced in 6h-fasted AHKO mice and did not rise following a glucose bolus as opposed to control mice (Fig. 6c), indicating that fasted middle-aged and aged AHKO mice were more insulin sensitive compared to age-matched control mice. Glucose clearance in response to glucose or insulin was unchanged in middle-aged AHKO mice (Fig. 6d) which—at that age—showed increased hepatic triglyceride accumulation. Interestingly, with advanced lipodystrophy but reversed fatty liver phenotype, at the age of 30–40 weeks, AHKO mice became more glucose tolerant and insulin sensitive compared to age-matched control mice (Fig. 6e). In accordance with the reversal of the liver phenotype, insulin-stimulated phosphorylation of AKT$^{pSer473}$ was higher in the liver of aged AHKO mice compared to control mice (Fig. 6f and Supplementary Fig. 4e). Thus, along with the amelioration of the

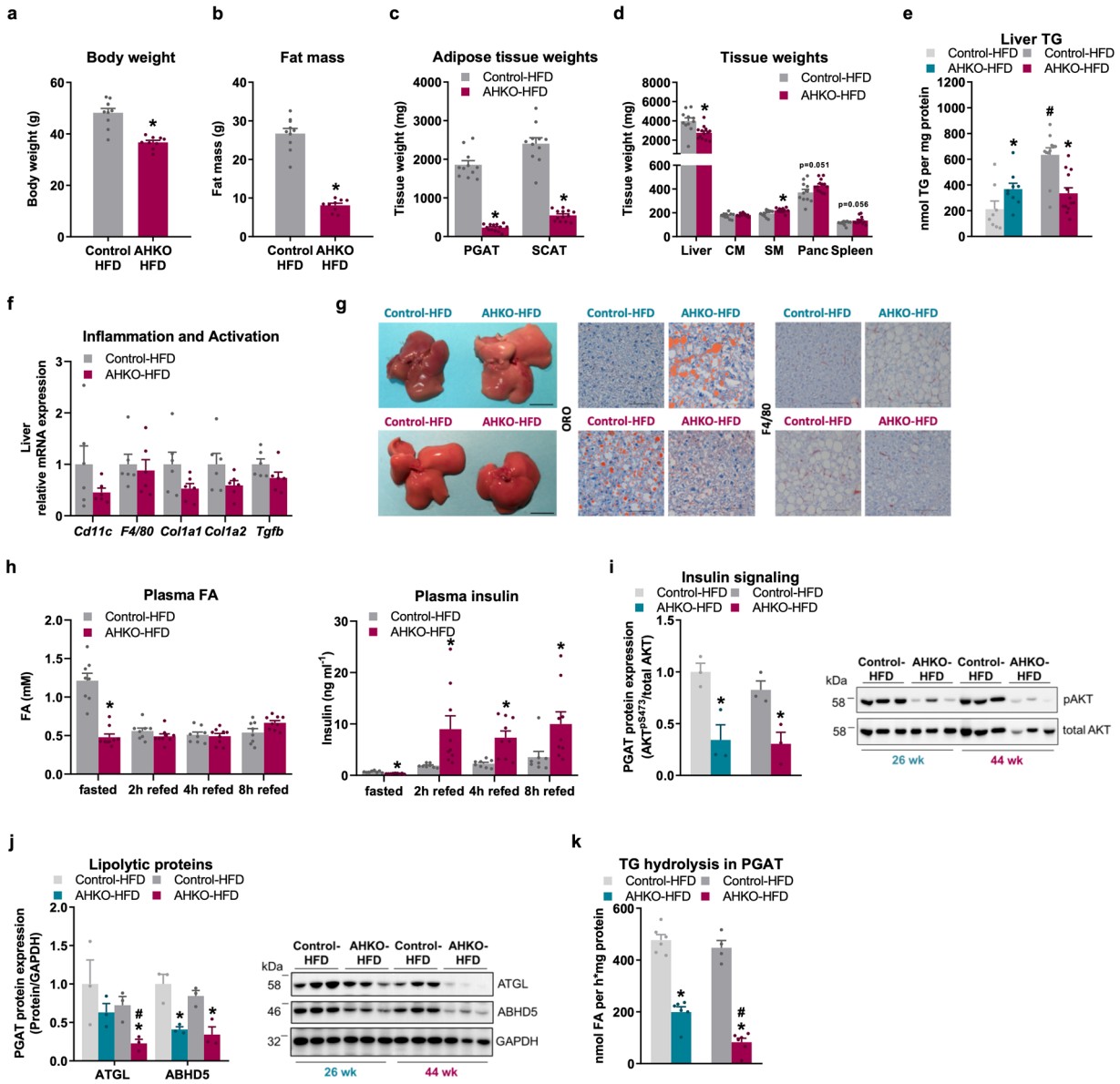

**Fig. 4 Effects of longitudinal adipocyte-specific HSL deletion on systemic and tissue-specific lipid metabolism. a** Body weight and **b** total fat mass (M, 44 weeks, HFD, ad libitum fed, $n = 9$–10 animals/genotype). **c** Adipose tissue weights (PGAT, SCAT) (M, 46 weeks, HFD, ad libitum fed, $n = 11$–13 animals/genotype). **d** Tissue weights of non-adipose tissues (CM cardiac muscle, SM skeletal muscle [quadriceps], Panc pancreas) (M, 46 weeks, HFD, ad libitum fed, $n = 11$–13 animals/genotype). **e** Liver TG content (M, 26 and 46 weeks, HFD, ad libitum fed, $n = 10$–13 animals/genotype). **f** Hepatic mRNA expression of immune cell markers and genes involved in fibrosis relative to *Cyclophilin* reference gene by qPCR with control animals arbitrarily set to 1 for each gene (M, 44 weeks, HFD, ad libitum fed, $n = 6$ animals/genotype). **g** Representative gross images (scale bar: 1 cm) and histological staining of liver sections with ORO (neutral lipids) and against F4/80-postive immune cells (scale bar: 100 μm; M, 26 and 46 weeks, HFD, ad libitum fed). **h** 12 h fasted and refed plasma FA (fatty acid; left) and insulin (right) levels (M, HFD, 38 weeks, $n = 6$–7 animals/genotype). **i** Insulin signaling in WAT. Mice were injected with insulin at 0.75 IU kg⁻¹ body weight (M, 26 and 44 weeks, HFD, 12 h fasted, $n = 3$ animals/genotype). Left: quantification of phosphorylation of AKT$^{pSer473}$/total AKT in PGAT. Right: representative immunoblots. **j** Left: PGAT protein expression of lipolytic proteins and regulator (ATGL, ABHD5). Right: representative immunoblots (M, 26 and 44 weeks, HFD, 2 h fed, $n = 3$ animals/genotype). **k** Triglyceride hydrolase activity in PGAT determined by [³H]-labeled oleic acid release (M, 26 and 44 weeks, HFD, 2 h fed, $n = 4$–6 animals/genotype). Data represent mean + SEM. Statistical significance was determined by Student's two-tailed *t* test. $P < 0.05$: * for effect of genotype; # for effect of age.

liver phenotype, hepatic insulin signaling and systemic glucose homeostasis were improved in aged AHKO mice.

## Discussion

Inhibition of adipocyte lipolysis has been demonstrated in numerous studies to counteract obesity and obesity-related disorders such as insulin resistance, type 2 diabetes, and non-alcoholic fatty liver disease[6,9,11]. Deregulated adipose tissue lipolysis is thought to be the primary cause for elevated plasma fatty acid levels, which is a strong single predictor for insulin resistance[20,21]. We and others have demonstrated that modulation of adipocyte lipolysis via genetic ablation or pharmacological

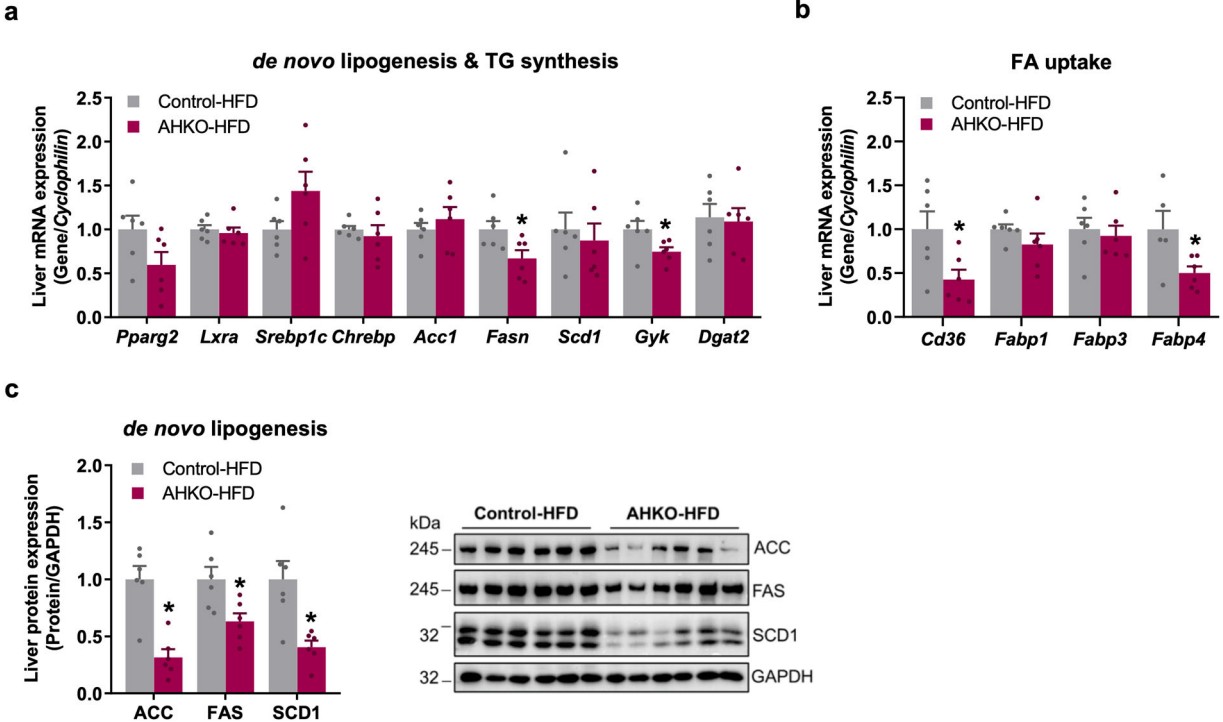

**Fig. 5 Effects of longitudinal adipocyte-specific HSL deletion on liver lipid metabolism. a** Hepatic mRNA expression of genes involved in de novo lipogenesis and triglyceride (TG) synthesis and **b** genes involved in fatty acid uptake relative to *Cyclophilin* reference gene by qPCR with control animals arbitrarily set to 1 for each gene (M, 44 weeks, HFD, ad libitum fed, $n = 6$ animals/genotype). **c** Left: expression of proteins involved in de novo lipogenesis (ACC, FAS, SCD1) in the liver. Right: representative immunoblots (M, 44 weeks, HFD, ad libitum fed, $n = 6$ animals/genotype). Data represent mean + SEM. Statistical significance was determined by Student's two-tailed t test. $P < 0.05$: * for effect of genotype.

inhibition of ATGL improves systemic glucose homeostasis and insulin sensitivity, and protects against hepatic steatosis in the setting of diet- or genetically induced obesity[6,7,9]. Consistently, HSL haplo-insufficient mice or mice treated with an HSL inhibitor exhibited increased glucose tolerance and insulin sensitivity as well as lower hepatic triglyceride levels[11,22]. However, a recent study on mice lacking HSL specifically in adipose tissue reported that, despite reduced plasma fatty acid levels, animals progressively develop insulin resistance and hepatic steatosis[12]. In accordance with global HSL-KO mice[15], mice lacking HSL specifically in the adipose tissue also developed a lipodystrophic phenotype.

In this study, we investigated the functional role of adipocyte HSL activity in whole-body energy and lipid homeostasis in the setting of diet-induced obesity. We employed AHKO mice as a genetic mouse model lacking functional HSL only in mature adipocytes and fed them HFD for up to 46 weeks to induce obesity. To delineate phenotypical changes over time, we characterized adipose tissue function and systemic metabolism of AHKO mice at young (10 weeks), middle (26 weeks), and advanced age (46 weeks). We found in the stage from young to middle-age that AHKO mice showed (1) less adipose tissue mass expansion accompanied by reduced expression of PPARg and its target genes involved in lipid metabolism as well as increased immune cell infiltration and inflammation in WAT; (2) impaired adipose tissue insulin signaling and insulin-mediated suppression of lipolysis during refeeding; and (3) greater hepatic triglyceride accumulation due to increased flux of fatty acids from adipose tissue upon refeeding and PPARg-induced expression of lipogenic genes (independent of SREBP1c). Remarkably, the fatty liver phenotype reversed with advanced age due to exacerbation of the lipodystrophic phenotype, which declined the lipolytic response

of the adipose tissue. Consequently, plasma fatty acid levels of aged AHKO mice remained entirely unchanged between fasting and refeeding. Advanced lipodystrophy and the reversed fatty liver phenotype in aged AHKO mice were accompanied by improved glucose homeostasis compared to age-matched obese control mice. Thus, this study demonstrates that inhibition of adipocyte HSL activity provokes adipocyte dysfunction, leading to tissue-specific insulin resistance, inflammation, and fat mass reduction. These progressive deteriorations of adipose tissue mass and function provoke fatty liver without affecting glucose homeostasis at this stage. When lipodystrophy advances, further blunting adipose tissue lipolysis, fatty liver reverses and glucose homeostasis improves. Thus, the absence of functional lipolysis by HSL, similar as by blocking the initial and rate-limiting step of lipolysis by ATGL or its coactivator ABHD5[6,9], leads to the loss of adipocyte plasticity and metabolic flexibility. Hence, energy supply depends on alternative energy fuel which goes along with increased systemic glucose clearance and improvement of an adverse liver phenotype.

In agreement with HSL-deficient humans[13] and previously reported atHSL-KO mice[12], HFD-fed AHKO mice also showed a progressive reduction in fat mass expansion. Energy balance analysis showed that AHKO mice had rather higher than lower food intake with age. Fecal energy output was not different, precluding the possibility that a reduced use of calories from the HFD accounts for the lean phenotype in AHKO mice. Furthermore, AHKO mice did not exhibit browning/beiging of WAT depots, excluding increased thermogenesis in WAT being causative for the reduced fat mass gain. Although we did not measure differences in total energy expenditure between the genotypes, we cannot exclude that the indirect calorimetry is not sensitive enough to detect small increases in energy expenditure[9,10,23,24],

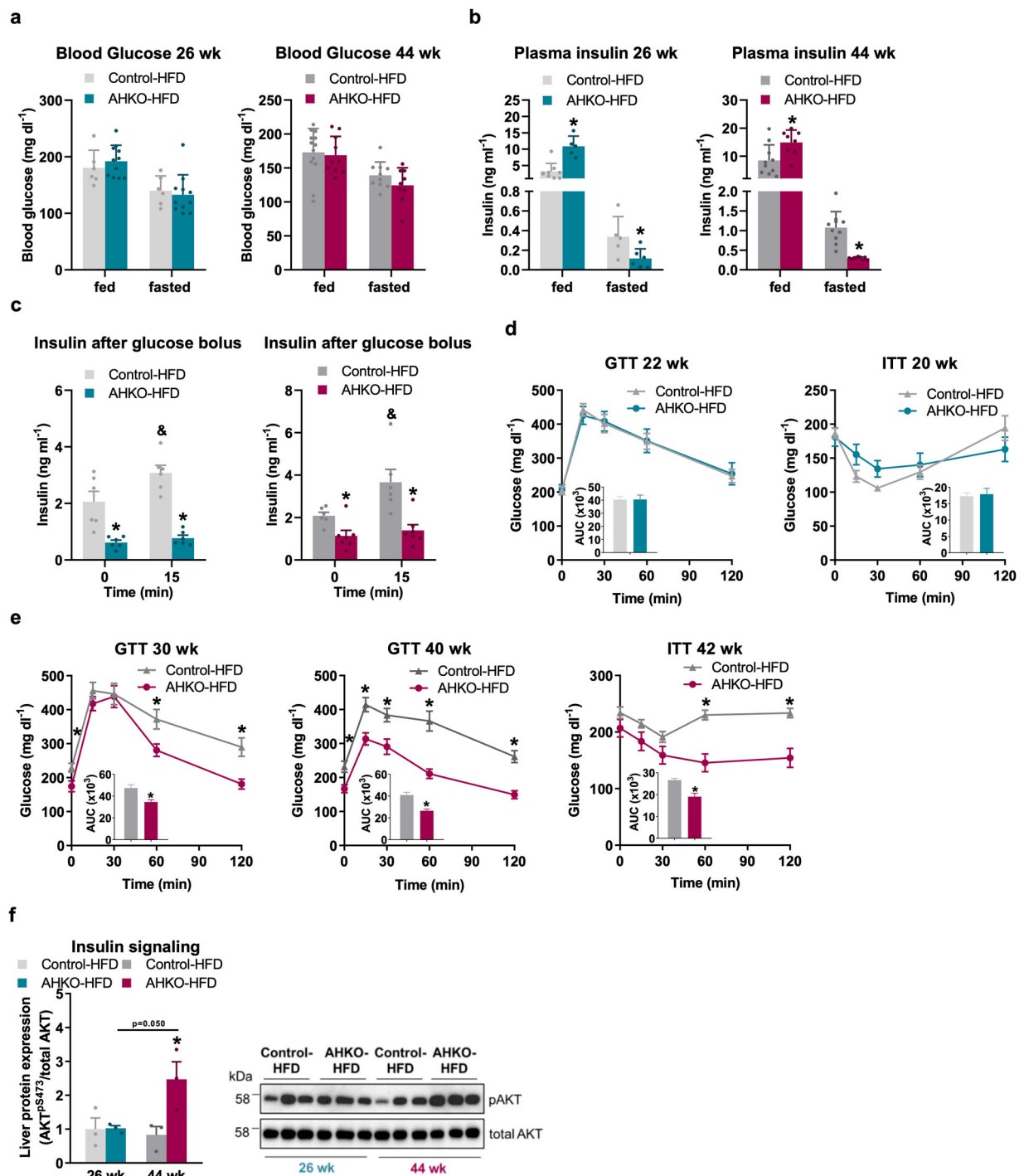

**Fig. 6 Effects of adipocyte-specific HSL deletion on systemic glucose homeostasis and insulin action. a** Ad libitum fed and 12 h fasted blood glucose (M, 26 and 44 weeks, HFD, $n = 5-9$ animals/genotype). **b** Ad libitum fed and 12h fasted plasma insulin (M, 26 and 44 weeks, HFD, $n = 5-9$ animals/genotype). **c** Plasma insulin levels following glucose bolus (1.6 g glucose kg$^{-1}$ body weight; M, 26 and 42 weeks, HFD, 6 h fasted, $n = 6-7$ animals/genotype). **d** Glucose tolerance test (GTT; left) with 1.6 g glucose kg$^{-1}$ body weight (M, 22 weeks, HFD, 6 h fasted, $n = 9-12$ animals/genotype). Insulin tolerance test (ITT; right) with 0.75 IU insulin kg$^{-1}$ body weight (M, 20 weeks, HFD, 4 h fasted, $n = 8-11$ animals/genotype). **e** GTT (left, middle) with 1.6 g glucose kg$^{-1}$ body weight (M, 30 and 40 weeks, HFD, 6 h fasted, $n = 9-12$ animals/genotype). ITT (right) with 0.75 IU insulin kg$^{-1}$ body weight (M, 42 weeks, HFD, 4 h fasted, $n = 9-10$ animals/genotype). **f** Insulin signaling in liver of middle-aged and aged control and AHKO mice. Mice were injected with 0.75 IU insulin kg$^{-1}$ body weight and tissues were collected after 15 min. Left: quantification of phosphorylation of AKT$^{pSer473}$/total AKT (M, 26 and 44 weeks, HFD, 12 h fasted, $n = 3$ animals/genotype). Right: a representative immunoblot. Data represent mean ± SEM. Statistical significance was determined by Student's two-tailed $t$ test. $P < 0.05$: * for effect of genotype; $^{\&}$ for effect of treatment.

which provoke the progressive loss of fat mass. The trend toward increased locomotor activity could hint toward rather increased energy expenditure as well. Similarly, *Holm* and coworkers have demonstrated that mice globally lacking HSL fed HFD exhibit increased energy expenditure[25].

The gradual development of lipodystrophy in AHKO mice was accompanied by a successive downregulation of target genes involved in lipo/adipogenesis, lipid uptake, and lipid synthesis in adipose tissue. These data support previous studies showing that lipolysis is intertwined with lipid synthesis and storage of the adipose tissue itself in an "autocrine" fashion, and therefore critical for maintenance of adipose tissue function[6,10,14,26].

Interestingly, the knockout of either ATGL[6,10] or HSL reduces the expression of PPARg and its target genes in WAT, but only the loss of adipocyte HSL resulted in a striking lipodystrophic phenotype. Thus, HSL not only functions in adipose tissue to provide fatty acids as signaling lipids through its triglyceride and diglyceride hydrolase activity[27], but also through its retinyl ester hydrolase activity in WAT[4]. As demonstrated by Ström et al.[4], the adipose tissue of mice globally lacking HSL have reduced levels of retinoic acid[4]. Consequently, the expression of genes known to be positively regulated by retinoic acid is downregulated including factors promoting adipocyte differentiation. More importantly, administration of retinoic acid partially reverses the lipodystrophic phenotype, demonstrating that at least in part disturbance of retinoic signaling is causative for impaired fat mass expansion in global HSL-KO mice.

This study demonstrates yet another mechanism by which HSL deficiency contributes to lipodystrophy: uncontrolled fatty acid release upon refeeding. Plasma fatty acids of AHKO mice remained high after refeeding since insulin signaling in adipose tissue was impaired and, thus, insulin-mediated suppression of lipolysis was defective. This uncontrolled fatty acid release from adipose tissue most likely contributed to the decline in adipose tissue mass in AHKO mice. Similar mechanisms are also causative for the lipodystrophic phenotype of perilipin 1-KO mice, where increased basal lipolytic activity causes uncontrolled and elevated fatty acid release over time[28]. Such uncontrolled fatty acid release might also stimulate the recruitment of immune cells in AHKO adipose tissue, since fatty acids released from adipocytes are known to induce immune cell infiltration[6,29,30]. On the other hand, dysfunctional adipocytes and their subsequent cell death might further promote an inflammatory response[19]. However, additional studies are required to assess the immune phenotype in AHKO mice. Thus, impaired adipocyte HSL action profoundly changes lipid metabolism, expansion capacity, and health of adipose tissue.

Despite the similarities with respect to the lipodystrophy and fatty liver phenotypes in AHKO mice in this study vs. in atHSL-KO mice in a previously reported study[12], there are notable differences related to both the underlying mechanism and metabolic consequences. We suggest that the development of the fatty liver phenotype depends on the uncontrolled release of fatty acids upon refeeding secondary to impaired insulin-mediated suppression of residual lipolysis, thereby channeling fatty acids from adipose tissue to the liver. Consequently, AHKO mice gradually accumulated triglycerides in the liver in early to mid-life. The fatty liver phenotype of AHKO mice reversed when plasma fatty acid levels normalized, which occurred in aged AHKO mice, that were devoid of adipose tissue lipolytic capacity. These observations indicate that deregulated fatty acid release from the insulin-resistant adipose tissue of middle-aged AHKO mice contributed to hepatic triglyceride accumulation.

Interestingly, the expression of hepatic *Pparg2* directly correlated with plasma fatty acid levels in AHKO mice. Consistently, it has been previously shown that ATGL and HSL activities are required to provide lipid metabolites that induce PPARg expression[10,27]. Thus, impaired suppression of adipocyte lipolysis and its accompanied release of fatty acids from adipose tissue might directly or indirectly activate hepatic PPARg but not adipose tissue PPARg. These opposite results are caused most likely by the dysfunctional state of the adipose tissue. In accordance with increased expression of hepatic *Pparg2* in middle-aged AHKO mice, induction of PPARg was shown to promote hepatic lipid accumulation, while knockdown of PPARg protects from hepatic steatosis[31,32]. Simultaneously to activation of *Pparg2*, expression of lipogenic enzymes is upregulated in mouse models with hepatic steatosis[33,34]. Correspondingly, expression of target genes involved in de novo lipogenesis was increased in the presence of high *Pparg2* expression levels, but unchanged or even reduced when *Pparg2* expression was lower in AHKO mice. Remarkably, the induction of lipogenic gene expression was independent of SREBP1c, the transcription factor thought to be the main regulator of de novo lipogenesis. This finding is surprising since SREBP1c activation is mostly regulated by insulin[35,36], and both middle-aged and aged AHKO mice had increased plasma insulin levels. However, a previous study demonstrated that *Pparg2* expression strongly induces de novo lipogenesis in mice with hepatic steatosis without altering SREBP1c activity[32]. Thus, in AHKO mice the fatty liver phenotype is further induced by increased de novo lipogenesis, presumably via adipocyte derived fatty acid-mediated PPARg2 activation.

A previous study reported that the fatty liver phenotype of atHSL-KO mice is associated with reduced beta oxidation and VLDL secretion[12]. Even though we did not measure differences in the expression of genes involved in beta oxidation in ad libitum fed liver of AHKO mice, we observed reduced fasting plasma ketone bodies, suggesting that hepatic ketogenesis was reduced in AHKO mice as shown in aged atHSL-KO mice. In accordance with impaired adipocyte lipolysis and reduced fatty acid release upon fasting in AHKO mice, inhibition of adipose tissue ATGL or its coactivator ABHD5 reduces expression of key enzymes regulating beta oxidation and ketogenesis as well. However, and in contrast to atHSL-KO mice[12], mice specifically lacking ATGL or ABHD5 in adipocytes do not accumulate triglycerides in the liver[6,17]. Notably, AHKO mice of this study showed reduced fasting plasma triglyceride levels, implying that VLDL release of these mice was decreased, an observation consistent with the finding of reduced VLDL secretion of atHSL-KO mice[12].

One of the most noteworthy observations of this study was that the fatty liver phenotype of AHKO mice reversed with advanced age, concurrently when lipodystrophy exacerbated, whereas glucose homeostasis improved. These changes occurred with normalization of plasma fatty acid levels upon refeeding which most likely resulted from a decline in adipocyte ATGL-mediated lipolytic activity. In support of this hypothesis, the protein levels of ATGL and its coactivator ABHD5 were reduced in adipose tissue of aged AHKO mice, and triglyceride hydrolytic activity was further declined in adipose tissue lysates of these mice in in vitro hydrolase activity assays. Despite this progressive deterioration in the lipodystrophic adipose tissue, but in accordance with our hypothesis, the lack of refed plasma fatty acid levels normalized the expression of PPARg2. Consequently, expression levels of key enzymes regulating de novo lipogenesis also normalized to levels of control mice. Over time, these changes likely disposed of the lipid load in the liver, which led to improvement of the fatty liver phenotype. Normalization of the fatty liver phenotype was also apparent from reduced expression of immune cell and fibrotic markers. Similarly,

insulin signaling improved in liver of aged AHKO mice, suggestive of increased hepatic glucose utilization. Indeed, direct assessment of glucose clearance by glucose tolerance test in aged AHKO mice showed improved glucose homeostasis. Thus, the beneficial impact on systemic glucose homeostasis upon prolonged inhibition of adipocyte HSL activity is partially mediated by blunted lipolytic capacity of the adipose tissue. Additional studies are required to specifically investigate mechanisms and factors such as hormones and signaling molecules that impact the interplay between adipose tissue and liver to regulate energy homeostasis.

In conclusion, our study underscores the importance of adipocyte HSL maintaining adipose tissue mass and function and its impact on hepatic and systemic metabolism during aging.

## Methods

**Animals**. AHKO mice were generated by crossing HSL-flox (HSL$^{flox/flox}$ mice[3] with mice expressing Cre recombinase under the control of the adiponectin promoter[37]. Mice were backcrossed onto C57BL/6J for >N10. HSL$^{flox/flox}$ Cre/+ (AHKO) and HSL$^{flox/flox}$+/+ (control) mice were used for experiments. All animal experiments were approved by the Austrian Federal Ministry for Science, Research, and Economy (protocol number BMWFW-66.007/0026/-WF/V/3b/2017) and the ethics committee of the University of Graz and were conducted in compliance with the council of Europe Convention (ETS 123). All studies involving animals are reported in accordance with the ARRIVE guidelines for reporting experiments involving animals[38]. Mice were housed under standard conditions (21–23 °C, 14:10 h light:dark cycle) with ad libitum access to chow diet (11 kJ% fat; V1126, Ssniff Spezialdiäten GmbH) or HFD (45 kJ% fat; E15744, Ssniff Spezialdiäten GmbH) in a specific pathogen free environment. HFD treatment started at 4–5 weeks of age. For all studies, male age-matched littermates were used as controls. Genotype, sex, age, and number are indicated for each experiment in the appropriate figure legends.

**Metabolic phenotyping**. Body mass composition was determined by nuclear magnetic resonance (NMR) spectroscopy using a TD-NMR miniSpec Live Mice Analyzer (Bruker Optics, Billerica, USA). For food intake monitoring, genotype-matched mice were housed together. For glucose tolerance test, mice were injected intraperitoneally (ip) with 1.6 g glucose per kg body weight following a 6 h fast. For insulin tolerance test, mice were injected ip with 0.75 IU per kg body weight following a 4 h fast. Blood glucose was determined using a Wellion Calla glucometer (MedTrust, Marz, Austria). For lipid tolerance test, mice received a gavage containing 250 µl of olive oil following a 4 h fast. Plasma lipid parameters were determined using the following commercial kits: fatty acid (HR Series NEFA-HR Reagents, Wako Diagnostics), glycerol (Free glycerol reagent, Sigma-Aldrich), triglyceride (Infinity Triglycerides Liquid Stable Reagent, ThermoFisher Scientific), and ketone bodies (Beta-Hydroxybutyrate, Cayman). Plasma insulin was determined using Ultra-Sensitive Mouse Insulin ELISA Kit (Crystal Chem). Plasma leptin was measured using Mouse Leptin ELISA Kit (Crystal Chem). Plasma ALT levels were determined using the Infinity ALT (GPT) Liquid stable reagent (ThermoFisher Scientific). To stimulate in vivo lipolysis, 6h-fasted mice were injected ip with 1 mg CL-316,243 (Sigma-Aldrich) per kg body weight and blood samples were collected prior and 15 min after treatment.

**Hepatic lipid content**. Neutral lipids were extracted from liver tissues (20–50 mg) according to Folch et al.[39]. Briefly, tissues were homogenized in 1 ml of chloroform/methanol (2/1, v/v) using a ball mill (Retsch GmbH). For phase separation, 200 µl of $H_2O$ were added, samples were vortexed for 30 s and centrifuged at 5000 × $g$ for 10 min at 4 °C. Lower organic phase was collected. For repeated extractions, 500 µl of chloroform were added, mixed by vortexing, and centrifuged as above. The lower organic phases were combined and dried in a speed-vac (Labconco). Extracts were dissolved in 0.1% Triton X-100 by incubation for 4 h at 37 °C at 500 rpm in a thermomixer (Eppendorf) and subsequently sonicated in a water bath sonicator (Transsonic T460, Elma). Triglyceride content was analyzed by commercial Infinity Triglycerides Liquid Stable Reagent kit. For protein determination, dried infranatant was dissolved in 0.1% SDS/0.3N NaOH for 4 h and protein content was determined by Pierce™ BCA Protein Assay Kit (ThermoFisher Scientific) according to manufacturer's instructions using BSA as standard.

**Lipolytic activities in adipose tissue and liver**. Adipose tissue (perigonadal; PGAT) and liver were homogenized in ice-cold solution A (0.25 M sucrose, 1 mM EDTA, 1 mM DTT supplemented with 20 µg ml$^{-1}$ leupeptin, 2 µg ml$^{-1}$ antipain, and 1 µg ml$^{-1}$ pepstatin; pH 7.0) using Ultra-Turrax Homogenizer (IKA). Adipose tissue homogenates were centrifuged at 20,000 × $g$ at 4 °C for 30 min. Liver homogenates were centrifuged at 1000 × $g$ at 4 °C for 10 min. The lipid-poor infranatant was used for measuring hydrolytic activities. Protein concentrations were determined by Bio-Rad protein assay (Bio-Rad) according to manufacturer's instructions using BSA as standard.

Triglyceride hydrolase activity assay was performed as described[40] with some modifications. The triglyceride substrate contained 300 µM triolein, 10 µCi per ml [$^3$H] triolein, and 45 µM PC/PI (3/1, M/M). Lipids were dried under N$_2$ and emulsified by sonication in 100 mM K-phosphate buffer (pH 7.4). Then, 5% fatty acid-free BSA was added. Twenty µg PGAT lysate or 90–150 µg liver lysate in 100 µl solution A were incubated with 100 µl substrate for 1 h at 37 °C. Reactions were terminated by addition of 3.25 ml methanol/chloroform/$n$-heptane (10/9/7, v/v/v) and 1 ml 100 mM K-carbonate (pH 10.5). Then, samples were vigorously vortexed and centrifuged at 2000 × $g$ for 10 min. The radioactivity in 500 µl of the upper phase was determined by liquid scintillation counting. Substrate blank incubation was performed with solution A.

For ex vivo lipolysis assay, PGAT explants from ad libitum fed mice were cut into small pieces and put into DMEM (Gibco, Invitrogen, ThermoFisher Scientific) supplemented with 2% fatty acid-free BSA for 60 min at 37 °C. The assay was performed in the presence (stimulated) or absence (basal) of 10 µM isoproterenol (Sigma-Aldrich)[40]. Glycerol release was measured using commercial kit Free glycerol reagent.

**Protein and gene expression**. To investigate protein expression in tissues, liver and adipose tissues were homogenized in ice-cold solution A using Ultra-Turrax Homogenizer. Then, homogenates were centrifuged for 10 min at 1000 × $g$ at 4 °C and protein concentrations were determined by Bio-Rad protein assay according to manufacturer's instructions using BSA as standard. Lysates (20–30 µg) were delipidated overnight at −20 °C using a fivefold volume of ice-cold acetone. Proteins were precipitated by centrifugation at 20,000 × $g$ at 4 °C for 30 min, dissolved in 1x SDS sample buffer, separated by SDS-PAGE (7.5–12.5% Tris-glycine), and transferred onto a PVDF membrane (Carl Roth GmbH). The membrane was blocked with 10% nonfat dry milk in TST (50 mM Tris-HCl, 0.15 M NaCl, 0.1% Tween-20; pH 7.4). Primary antibodies used for protein expression analysis are listed in Supplementary Table 1. Immunoblots were visualized by enhanced chemiluminescence using Clarity Western ECL plus Western Blotting Detection Reagent (ThermoFisher Scientific) and ChemiDoc Touch Imaging System (Bio-Rad). Signal intensities were quantified by densitometric analyses using Image Lab software (version 5.2.1; Bio-Rad). Full-sized images of Western blots are summarized in Supplementary Figs. 5–7.

For gene expression, adipose tissue (~100 mg) and liver (~30 mg) were homogenized in 1 ml TRIzol using Ultra-Turrax Homogenizer and incubated at room temperature for 5 min. Phase separation was performed by addition of 100 µl 1-bromo-3-chloropropane and centrifuged at 12,000 × $g$ and 4 °C for 15 min. For adipose tissue, the centrifugation step was repeated twice to remove the entire fat cake. Clear supernatant was transferred and total RNA was precipitated by addition of 500 µl isopropyl alcohol and centrifuged at 12,000 × $g$ and 4 °C for 15 min. DNase-digested RNA was reverse-transcribed into cDNA using transverse transcriptase kit (Quiagen). PCR was performed using StepOnePlus Real-time PCR system (Applied Biosystem). Briefly, PCR reaction contained 8 ng cDNA, 10 pmol forward/reverse primer, and 10 µl iTaq™ Universal SYBR® Green Supermix (Bio-Rad). PCR denaturation temperature was 95 °C and annealing/extension was performed at 60 °C over 40 cycles[17]. Primer sequences are listed in Supplementary Table 2. Target gene expression was calculated by the $\Delta\Delta Ct$ method and normalized to Cyclophilin B as reference gene.

**Insulin signaling in tissue**. Evaluation of insulin signaling was performed as described previously[6]. Briefly, 12h-fasted mice were injected ip with saline or 0.75 IU insulin per kg body weight. Ten minutes thereafter, mice were sacrificed and tissues were snap frozen in liquid nitrogen. Tissues were homogenized in ice-cold lysis buffer (50 mM Tris-HCl, 150 mM NaCl, 1 mM EDTA, 1% NP-40, 20 µg ml$^{-1}$ leupeptin, 2 µg ml$^{-1}$ antipain, 1 µg ml$^{-1}$ pepstatin, and phosphatase inhibitor PhosStop [Roche]; pH 7.5) using Ultra-Turrax Homogenizer. Then, samples were centrifuged at 20,000 × $g$ at 4 °C for 30 min and supernatant excluding floating fat layer was collected using a syringe. Western blot analysis was performed as described above. Primary antibodies used for protein expression analysis are listed in Supplementary Table 1.

**Histological analysis**. For immune cell staining, liver was fixed in 4% neutral-buffered formalin and embedded in paraffin (Tissue Tek Tec, Sakura). Samples were sectioned (2 µm) and incubated with anti-mouse F4/80 antibody (1:50; Serotec MCA 497GA, Bio-Rad) and counterstained with Mayer's hematoxylin for nuclei using standard histological techniques. For neutral lipid staining, fixed liver was embedded in OCT-Combound (Sakura), sectioned (4 µm), and stained using Oil Red O (ORO) and counterstained with Mayer's hematoxylin for nuclei using standard histological techniques.

**Statistical analysis and reproducibility**. Results are expressed as mean ± SEM. Statistical analysis was performed using GraphPad Prism 8 (GraphPad Software, San Diego). The exact numbers of replicates are presented in individual figure

legends. Comparisons were made by unpaired two-tailed Student's *t* test. *P* values of <0.05 were considered statistically significant.

**Reporting summary**. Further information on research design is available in the Nature Research Reporting Summary linked to this article.

## Data availability
All raw data analyzed in this paper are available in the Supplementary information and Supplementary Data 1. Any remaining information can be obtained from the corresponding author upon reasonable request.

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

## Acknowledgements
This work was supported by the following: DOC fellowship 25049 (to L.P.) funded by the Austrian Academy of Science (OEAW); Stand-alone project-P28882-B21 (to G.S.), P31638-B34 (to U.T.), and P32225-B (to F.P.W.R.) as well as by SFB F73 (to G. Haemmerle, G.S., R.S., U.T.) and DACH I3535 (to A.L.) funded by Austrian Science Fund (FWF); Flagship Project "Lipases and Lipid Signaling" (to A.L.) funded by BioTechMed-Graz. We acknowledge the support of the field of excellence BioHealth and NAWI Graz. The authors especially thank Rudolf Zechner for the critical discussions and excellent input. We thank Kathrin A. Zierler, Astrid Steiner, and Birgit Juritsch for animal care and genotyping, Julia Stermscheg and Wolfgang Krispel for technical assistance, Silvia Schauer for histological staining, and Helga Reischl for assisting with bomb calorimetry measurement.

## Author contributions
L.P., U.T., and G.S. were the project leaders and contributed to all aspects of this work. A.L., E.E.K., M. Schweiger, T.O.E., and G. Haemmerle provided intellectual expertise. A.T., P.H., P.K., S.K., F.P.W.R., I.P., C.W., M. Schratter, S.E., S.H., and R.S. performed experiments. G. Hoefler was responsible for histological analysis. All authors contributed intellectually to this work and reviewed and edited the paper. G.S. is the guarantor of this work and, as such, has full access to all the data in the study and takes responsibility for the integrity of the data and the accuracy of the data analysis.

## Competing interests

The authors declare no competing interests.
