## [Peer Review File · Communications Biology]

Reviewers' comments:

Reviewer #1 (Remarks to the Author):

In the manuscript by Pjed, Taschler et al., the authors present somewhat interesting data on the short-term and long-term metabolic phenotype of deletion of hormone sensitive lipase in adipocytes. Although the lipodystrophic and fatty liver phenotype of adipocyte deletion of HSL has been previously described (Xia et al., Plos Genetics 2017), the current manuscript expands on the role of adipocyte HSL. Mice deficient in adipocyte HSL on a HFD become lipodystrophic by 26 weeks of age with fatty liver, effects which are attributed to reduced insulin suppression of adipocyte lipolysis. Interestingly by 44 wks of age the fatty liver and elevated ad libitum plasma fatty acids are reversed in mice deficient in adipocyte HSL. In addition, by 40 wks of age, HSL adipocyte deficient mice become more glucose and insulin sensitive. The manuscript is well written, data is clearly presented and appropriate use of statistics.

There are a few major issues that require attention:

1. The authors attribute the majority of the fatty liver phenotype in mice deficient in adipocyte HSL due to the inability of insulin to suppress lipolysis and subsequent increased fatty acid release. It seems somewhat counterintuitive that in the absence of HSL greater lipolysis would drive a fatty liver. While the evidence is convincing for defective insulin signaling in adipose tissue, can the authors conclusively attribute the phenotype to greater lipolysis? In other words, can the authors exclude reduced fatty acid uptake in adipose tissue as a mechanism? Without tracer kinetics it is difficult to conclusively determine whether the increase in FA is due to lipolysis or reduced fatty acid uptake into adipose tissue. It seems plausible that the fatty liver is due to lipodystrophy and impaired fatty acid sequestration of adipose tissue.

2. The authors argue that in middle aged mice on HFD insulin resistance at the level of lipolysis is driving the fatty liver and that in aged mice (44 wks) lipolysis is reduced due to decreased levels of ATGL and ABHD5 protein and TG hydrolase in PGAT. However, were levels of ATGL and ABHD5 and TG hydrolase activity measured at middle age and was insulin signaling measured in old age mice? It would help strengthen the authors arguments to have the measurements of insulin signaling and ABHD5-ATGL levels and hydrolase activity in middle and old aged mice side by side.

Minor:

What WAT depot was measured in Figure 3D?

If the long-term effect (ie 44 wks) of HSL deletion in adipocytes is due to blunted lipolysis wouldn't there be an expected rebound in PGAT and SCAT weight?

In addition to AKT signaling in Figure 3D, have the authors measured cAMP levels or another readout of PKA activity?

Page 5, line 73 what is meant by "progressive lipodystrophy shut of residual lipolytic capacity"

Page 5, line 73 what is the evidence for "lessening the refeeding effect"

The authors excluded reductions in food intake (Figure 2E) for changes in body weight. Can any of the effects on body weight be attributed to reduced energy expenditure?

Reviewer #2 (Remarks to the Author):

This manuscript by Pajed et al investigates the role of adipocyte HSL in whole body energy homeostasis. They show that adipocyte-specific HSL Knockout (AHKO) mice fed HFD develop fatty liver associated with enhanced de novo lipogenesis gene expression in liver. Interestingly, the fatty liver phenotype reverses in aged AHKO mice, possibly due to blunted lipolytic activity in adipose tissue with pronounced lipodystrophy.

In many ways this paper mirrors one from a few years ago by Xia et al (PLoS Genetics) that also found that AHKO mice develop lipodystrophy and fatty liver. Many of the first several figures in the current manuscript recapitulate that earlier data pretty faithfully. I think it's appropriate to publish a second paper on the subject, if only because many KO phenotypes don't replicate well, so having

this paper will make readers more comfortable that the results seen by Xia are reproducible and robust. That said, I think there also needs to be something additional in the second paper on the subject, and for this paper that is the remarkable observation that steatosis disappears with age (the Xia paper looked at AHKO mice up to 32 weeks of age—this paper studies mice at 44 weeks. Along with the reversal of fatty liver, the AHKO mice become more insulin sensitive at 44 weeks. Simply put, they transition from lipodystrophic (unhealthy) to lean (healthy). As written, however, the paper is very descriptive, and fails to provide any true mechanistic insight into this extremely interesting phenotype.

Specific concerns:

1. Where are the calories going? Are the 44 wk old AHKO mice eating less? Losing more calories in their feces? Burning more? This is the glaring gap in the story, I believe. Careful measurements of energy homeostasis (food intake, fecal bomb calorimetry, indirect calorimetry for OCR and EE) need to be performed before and after the phenotype switches from lipodystrophic to lean.
2. One possibility is that thermogenic adipocytes begin to dominate between 26 and 44 weeks, either brown or beige. Please assess this using QPCR of thermogenic mRNAs, UCP1 Western, adipose histology, etc...
3. Can the authors suggest a mechanism by which PPAR γ would be simultaneously down-regulated in WAT (Fig. 2) but up-regulated in liver (Fig. 3)?

Reviewer #3 (Remarks to the Author):

Pajed et al. provide an complete and complex manuscript. Finding a 'lipodystrophic' phenotype in AHKO mice on a HFD with an observation pointing towards regression of a fatty liver phenotype. The authors provide in depth analyses on the phenotype and provide an interesting conclusion. There are some questions that could be addressed to provide an insight.

Major:

- 1) The HFD was provided from weaning onwards. While food intake was not different during the 6 months of feeding in total kcal's how was the feeding pattern during the day?
- 2) Related to 1), if they eat the same amount but weight less where is the energy going? How about Energy Expenditure? How about fecal loss of energy? Please provide more details on this energy balance.
- 3) The liver phenotype of these AHKO mice on a HFD is interesting. Did the authors consider histology to confirm a pathological picture of the increased immune cell and activation?
- 4) The conclusion of remission (abstract) or regression (results/conclusion) seems a little far fetched? If you take a closer look at the TG accumulation you could conclude that there was no difference at 44wks of age, Fig4, D and E. And one other observation is that TG content increases 2-4-fold between 24wks and 44wks. What is the accumulation is progression more rapid in the AHKO mice and reaches a maximum somewhere between 24-44wks? Controls get there later? So no remission or regression but simple reaching maximal capacity without further increase possibility? A histological picture of the livers would provide a better insight as well.
- 5) The conclusion of the data in fig6 might require a different one too? Looking at the glucose graphs during the GGT it looks like there is better clearance but raises a couple of questions: a) the dose was per kg body weight and the AHKO mice are lighter compared to the controls. Did the authors do a GGT with an equal dose of glucose? b) secondly, based on the fed data the insulin response after a challenge seems higher, so please provide the insulin data during the GTT's. If there is more insulin during a GTT and a real increased glucose clearance this indicates either insulin resistance or insulin sensitivity.
- 6) Related to 5), the pAKT data is not provided proof for more insulin sensitivity, fig6. How do the basal pAKT levels look like? Please provide the increase compared to unstimulated tissue. For true IR or IS measurements consider the euglycemic hyperinsulinemic clamps.

Minor

1) Some of the data is provided from female mice and other from male mice? Is there any indication that there is no sex-difference?

Ms number/ID: COMMSBIO-20-0840-A

Point-to-Point Response to Reviewers' Comments

We thank the reviewers for their constructive comments. We have addressed and discussed all raised questions and comments. Please find our detailed response below.

Reviewers' comments:

Reviewer #1 (Remarks to the Author):

In the manuscript by Pjed, Taschler et al., the authors present somewhat interesting data on the short-term and long-term metabolic phenotype of deletion of hormone sensitive lipase in adipocytes. Although the lipodystrophic and fatty liver phenotype of adipocyte deletion of HSL has been previously described (Xia et al., Plos Genetics 2017), the current manuscript expands on the role of adipocyte HSL. Mice deficient in adipocyte HSL on a HFD become lipodystrophic by 26 weeks of age with fatty liver, effects which are attributed to reduced insulin suppression of adipocyte lipolysis. Interestingly by 44 wks of age the fatty liver and elevated adiposity plasma fatty acids are reversed in mice deficient in adipocyte HSL. In addition, by 40 wks of age, HSL adipocyte deficient mice become more glucose and insulin sensitive. The manuscript is well written, data is clearly presented and appropriate use of statistics.

The are a few major issues that require attention:

1. The authors attribute the majority of the fatty liver phenotype in mice deficient in adipocyte HSL due the inability of insulin to suppress lipolysis and subsequent increased fatty acid release. It seems somewhat counterintuitive that in the absence of HSL greater lipolysis would drive a fatty liver. While the evidence is convincing for defective insulin signaling in adipose tissue, can the authors conclusively attribute the phenotype to greater lipolysis? In other words, can the authors exclude reduced fatty acid uptake in adipose tissue as a mechanism? Without tracer kinetics it is difficult to conclusively determine whether the increase in FA is due to lipolysis or reduced fatty acid uptake into adipose tissue. It seems plausible that the fatty liver is due to lipodystrophy and impaired fatty acid sequestration of adipose tissue.

We thank the reviewer to bring this very important aspect to our attention. We have carefully reevaluated the possibility that impaired fatty acid uptake into adipose tissue drives the development of a fatty liver. We performed lipid tolerance test (LTT) using an olive oil gavage (**Suppl. Figure 2A; Fig. 1A** see below). The LTT of middle-aged mice demonstrated that systemic lipid uptake was comparable between AHKO and control mice. Interestingly, lipid clearance of aged AHKO mice increased as compared to controls (**Fig. 1A see below**), indicating that depending on age AHKO mice have either unchanged or improved systemic lipid clearance. To investigate adipose tissue-specific lipid uptake, we subjected mice to a [³H]-labeled triolein-supplemented olive oil gavage and determined lipid uptake. This gavage-experiment revealed no differences in adipose tissue-specific lipid uptake (**Suppl. Figure 2B**). We also treated mice with Bromo-palmitate (**Fig. 1B see below**), a non-metabolizable fatty acid analog, which allows to determine FA uptake directly. In agreement with the other lipid uptake studies, AHKO mice showed no differences in adipose tissue-specific FA uptake in both depots. These data support our hypothesis that rather uncontrolled lipolysis upon refeeding than impaired lipid uptake into adipose tissue drives the development of fatty liver in middle-aged AHKO mice. In agreement, we show that aged AHKO mice do not exhibit higher plasma FAs in the refeed state (**Figure 4H**), despite increased food intake, further supporting our hypothesis that a decline in adipocyte lipolysis results in the improvement of the fatty liver phenotype in AHKO mice.

Of note, we are aware that the performed experiments are not equal to tracer kinetic studies. However, our data strongly suggest that AHKO mice do not have impaired adipose tissue lipid uptake.

Fig 1: Systemic lipid clearance and adipose tissue-specific FA uptake in AHKO mice. **A.** Lipid tolerance tests (LTT) in middle-aged (left) and aged (right) mice. LTT were performed by oral olive oil gavage and subsequent measurement of plasma TGs (M, 26 wk & 42 wk, HFD, 4h-fasted, n=5-9/group). **B.** *In vivo* FA uptake into WAT using [¹⁴C]-labeled Bromo-palmitate (BP). BP was injected via the retro-orbital sinus and after 10 min tissue was collected (M, 20wk, HFD, 12h-fasted, n=4-5/group). Data represent mean + SEM. Statistical significance was determined by Student's two-tailed t-test. $P < 0.05$; * for effect of genotype.

2. The authors argue that in middle aged mice on HFD insulin resistance at the level of lipolysis is driving the fatty liver and that in aged mice (44 wks) lipolysis is reduced due to decreased levels of ATGL and ABHD5 protein and TG hydrolase in PGAT. However, were levels of ATGL and ABHD5 and TG hydrolase activity measured at middle age and was insulin signaling measured in old age mice? It would help strengthen the authors arguments to have the measurements of insulin signaling and ABHD5-ATGL levels and hydrolase activity in middle and old aged mice side by side.

Thank you for this constructive comment. We agree that an age-dependent comparison would strengthen our hypothesis. Therefore, we have now included Western Blot analyses of middle-aged and aged PGAT of HFD-fed mice in the revised manuscript (**Figure 4J**). We observed that ABHD5 protein expression was reduced in middle-aged and aged AHKO PGAT compared to age-matched control mice. Furthermore, ATGL protein expression was reduced in middle-aged AHKO PGAT and even more so in aged AHKO PGAT. Accordingly, TG hydrolase activity in WAT was decreased in PGAT of middle-aged mice and further reduced in aged AHKO mice (**Figure 4K**). In contrast to the age-dependent decline of lipolytic activity in WAT of aged AHKO mice, insulin signaling did not further reduce from middle-aged and aged AHKO mice (**Figure 4I**; **Suppl. Figure 3E**), indicating that WAT of both middle-aged and aged AHKO mice is similarly insulin resistant. Thus, the normalization of plasma FA levels of refeed aged AHKO mice that levels of aged control mice were not a consequence of recovered insulin-mediated suppression of lipolysis upon refeeding, but deterioration of lipolytic activity. Together, these data support our conclusion that - with age - the adipose tissue lipolytic function significantly reduces in AHKO mice, which consequently leads to amelioration of the fatty liver phenotype and systemic glucose homeostasis.

Minor:

What WAT depot was measured in Figure 3D?

We are sorry for the lack of clarity. We now stated in the text (line 178) that insulin signaling was determined in perigonadal and subcutaneous adipose tissue (PGAT and SCAT, respectively).

If the long-term effect (ie 44 wks) of HSL deletion in adipocytes is due to blunted lipolysis wouldn't there be an expected rebound in PGAT and SCAT weight?

That is a very good question. As shown in mouse models with defective lipolysis (i.e. global HSL-KO mice; global or adipocyte-specific ATGL-KO mice), the loss of lipases does not directly result in increased adipose tissue mass, especially when fed HFD (1, 2). Furthermore, studies in adipocyte-

specific ATGL-ko mice demonstrated that adipose tissue functions (i.e. lipolysis, lipid synthesis and storage) are intertwined and depend on each other (3). In line, we have made similar observations in the adipocyte-specific HSL-KO mice of this study, which showed reduced expression of target genes involved in lipid metabolism (**Figure 2H**). Consequently, blunted lipolysis is accompanied by reduced lipid synthesis and storage, apparently preventing a rebound in WAT weight.

In addition to AKT signaling in Figure 3D, have the authors measured cAMP levels or another readout of PKA activity?

We have measured PKA substrate signaling in WAT depots of middle-aged and aged mice (**Fig. 2** see below). The analyses showed that in aged AHKO mice the PKA substrate signaling was rather reduced, indicating that the PKA signaling cascade (including phosphorylation of lipid droplet-associated proteins such as PLIN1) progressively declines in AHKO mice.

To avoid excessive data presentation, we have not included these data in the manuscript.

Fig 2: Western Blot analysis in WAT depots of AHKO mice aged 24 wk and 46 wk. pPKA substrate in SCAT (left) and PGAT (right) using pPKA Substrate (RRXS/T) antibody (Cell signaling, #9624; 1:1,000 dilution).

Page 5, line 73 what is meant by “progressive lipodystrophy shut of residual lipolytic capacity”

We now have reworded the statement to read: Line 74-76: “*..our data show that progressive lipodystrophy is accompanied by reduced adipocyte lipolytic capacity in AHKO mice, thereby lowering the uncontrolled FA release upon refeeding, and improving glucose homeostasis.*”

Page 5, line 73 what is the evidence for “lessening the refeeding effect”

The evidence of the “refeeding effect” are increased plasma FA levels in middle-aged AHKO mice (**Figure 3C**). We hypothesize that this is caused by the lack of insulin-mediated suppression of adipose tissue lipolysis, a consequence of its insulin resistance. Insulin resistance of middle-aged and aged AHKO mice is evident from greatly reduced insulin signaling of the adipose tissue (**Figure 3D and 4I**) and elevated plasma insulin levels (**Figure 3C and 4H**). However, despite insulin resistance of the adipose tissue, aged AHKO mice showed no “uncontrolled” FA release from adipose tissue, most likely due to a further decline of adipose lipolytic activity (**Figure 4K**). This decline of adipose lipolytic activity apparently is “lessening the refeeding effect”, so that plasma FA levels normalize to control levels upon refeeding (**Figure 4H**).

To make our statement clearer, we have rephrased the sentence (line 74- 76) to read: “*Mechanistically, our data show that progressive lipodystrophy declines adipocyte lipolytic capacity in AHKO mice, thereby lowering the uncontrolled FA release upon refeeding, and improving glucose homeostasis.*”

The authors excluded reductions in food intake (Figure 2E) for changes in body weight. Can any of the effects on body weight be attributed to reduced energy expenditure?

We have also speculated that other reasons such as changes in energy expenditure (EE) could contribute to the phenotype. To get an overview of the systemic metabolic phenotype of AHKO mice, we monitored middle-age and aged mice in a laboratory animal monitoring system (PhenoMaster, TSE systems). Interestingly, we did not find differences in total energy expenditure (**Suppl. Figure 1A & 3A**). Because body weight is altered upon HFD and at different ages, we performed ANCOVA analyses using body weight as a co-variant. At an adjusted hypothetical body weight for AHKO and controls, the genotypes have comparable EE (**Fig. 3** see below). Additionally, food intake and fecal energy output were unchanged as well, excluding differences in caloric input or output as a cause for the observed lean phenotype in AHKO mice. At this time, we can just speculate that the indirect calorimetric measurement might be not sensitive enough to detect subtle changes in our AHKO mice (1, 4, 5, 6). Accordingly, small, but undetectable increases in EE in AHKO mice might be indicated by the longsome progression of lipodystrophy.

Fig 3. Energy expenditure (EE) in young, middle-aged, and aged mice. A. EE is expressed as adjusted means based on a normalized mouse weight 25.98 g determined using ANCOVA, $P=0.271$. (M, HFD, *ad libitum* fed, $n=5-6$ /group). B. EE is expressed as adjusted means based on a normalized mouse weight 35.41 g determined using ANCOVA, $P=0.654$. (M, HFD, *ad libitum* fed, $n=5-7$ /group). C. EE is expressed as adjusted means based on a normalized mouse weight 43.52 g determined using ANCOVA, $P=0.931$. (M, HFD, *ad libitum* fed, $n=12$ /group). Data represent mean + standard deviation.

References

- 1) Hypophagia and metabolic adaptations in mice with defective ATGL-mediated lipolysis cause resistance to HFD-induced obesity. *Schreiber R, et al. PNAS. 2015*. PMID: 26508640
- 2) Resistance to high-fat diet-induced obesity and altered expression of adipose-specific genes in HSL-deficient mice. *Kenji Harada, et al. Am J Physiol Endocrinol Metab. 2003* PMID: 12954598
- 3) Coupling of lipolysis and de novo lipogenesis in brown, beige, and white adipose tissues during chronic $\beta 3$ -adrenergic receptor activation. *Mottillo EP, et al. JLR. 2014*. PMID: 25193997
- 4) Pharmacological inhibition of adipose triglyceride lipase corrects high-fat diet-induced insulin resistance and hepatosteatosis in mice. *Schweiger M, et al. Nat Commun. 2017* PMID: 28327588
- 5) A recurring problem with the analysis of energy expenditure in genetic models expressing lean and obese phenotypes. *Butler AA, et al. Diabetes 2010* PMID: 20103710
- 6) A guide to analysis of mouse energy metabolism. Tschöp MH, et al. *Nat Methods. 2011*. PMID: 22205519 Free PMC article.

Reviewer #2 (Remarks to the Author):

This manuscript by Pajed et al investigates the role of adipocyte HSL in whole body energy homeostasis. They show that adipocyte-specific HSL Knockout (AHKO) mice fed HFD develop fatty liver associated with enhanced de novo lipogenesis gene expression in liver. Interestingly, the fatty liver phenotype reverses in aged AHKO mice, possibly due to blunted lipolytic activity in adipose tissue with pronounced lipodystrophy.

In many ways this paper mirrors one from a few years ago by Xia et al (PLoS Genetics) that also found that AHKO mice develop lipodystrophy and fatty liver. Many of the first several figures in the current manuscript recapitulate that earlier data pretty faithfully. I think it's appropriate to publish a second paper on the subject, if only because many KO phenotypes don't replicate well, so having this paper will make readers more comfortable that the results seen by Xia are reproducible and robust. That said, I think there also needs to be something additional in the second paper on the subject, and for this paper that is the remarkable observation that steatosis disappears with age (the Xia paper looked at AHKO mice up to 32 weeks of age—this paper studies mice at 44 weeks. Along with the reversal of fatty liver, the AHKO mice become more insulin sensitive at 44 weeks. Simply put, they transition from lipodystrophic (unhealthy) to lean (healthy). As written, however, the paper is very descriptive, and fails to provide any true mechanistic insight into this extremely interesting phenotype.

We thank the reviewer for his/her appreciation of our work. We agree that some of our data confirm previously published data on a similar mouse model. Despite a certain overlap, there are important differences to the work by Xia et al.: i) cleaner adipocyte-specific HSL deletion using AdipoQ-Cre mice; ii) our study investigates the short- and long-term effect of adipocyte-specific HSL deletion in the context of diet-induced obesity; and iii) we uncover a new mechanism how AHKO mice develop a fatty liver (uncontrolled FA release upon refeeding via impaired insulin-mediated suppression of lipolysis) and how with progression of lipodystrophy lipolytic capacity declines and uncontrolled FA release blunts thereby improving fatty liver phenotype in aged AHKO mice.

Specific concerns:

1. Where are the calories going? Are the 44 wk old AHKO mice eating less? Losing more calories in their feces? Burning more? This is the glaring gap in the story, I believe. Careful measurements of energy homeostasis (food intake, fecal bomb calorimetry, indirect calorimetry for OCR and EE) need to be performed before and after the phenotype switches from lipodystrophic to lean.

We agree with the reviewer that it is very important to investigate the fate of energy in AHKO mice in more detail. Therefore, we have monitored middle-aged and aged mice in a laboratory animal monitoring system (PhenoMaster, TSE systems). We have now included oxygen consumption, locomotor activity, food intake, and energy expenditure (EE) of middle-aged and aged mice into the manuscript (**Suppl. Figure 1A-D; Suppl. Figure 3A-D**). Middle-aged control and AHKO mice had comparable food intake, oxygen consumption, and locomotor activity. In contrast, aged AHKO mice had reduced oxygen consumption rate in the light phase but were more active and had higher food intake in the dark phase.

Middle-aged and aged mice showed no differences in EE. Because body weight impact on EE, we performed ANCOVA analyses using body weight as a co-variant (1). When adjusted to a hypothetical body weight, the genotypes had comparable EE (**Fig. 1** see below). However, we expect that AHKO mice have slightly increased energy expenditure – which we cannot detect with the indirect calorimetric measurement – and these subtle changes contribute to a slow-acting progression of the lipodystrophy phenotype in AHKO mice (1, 2, 3, 4).

Fig 1. Energy expenditure (EE) in young, middle-aged, and aged mice. **A.** EE is expressed as adjusted means based on a normalized mouse weight 25.98 g determined using ANCOVA, $P=0.271$. (M, HFD, *ad libitum* fed, $n=5-6$ /group). **B.** EE is expressed as adjusted means based on a normalized mouse weight 35.41 g determined using ANCOVA, $P=0.654$. (M, HFD, *ad libitum* fed, $n=5-7$ /group). **C.** EE is expressed as adjusted means based on a normalized mouse weight 43.52 g determined using ANCOVA, $P=0.931$. (M, HFD, *ad libitum* fed, $n=12$ /group). Data represent mean + standard deviation.

Furthermore, we have evaluated whether impaired intestinal food absorption in AHKO mice might contribute to the reduced body weight gain. Thus, we exemplarily determined feces mass output and fecal energy content of middle-aged mice. Both feces mass and its corresponding energy content were comparable between control and AHKO mice (**Suppl. Figure 1E**). This suggests that AHKO mice on HFD do not develop mal-absorption.

Overall, these data suggest that the phenotype of AHKO mice is provoked by a small, but undetectable increase in EE and not by changes in energy uptake or output.

2. One possibility is that thermogenic adipocytes begin to dominate between 26 and 44 weeks, either brown or beige. Please assess this using QPCR of thermogenic mRNAs, UCP1 Western, adipose histology, etc...

We found this comment very interesting and valuable. We have investigated the thermogenic phenotype in WAT of middle-aged AHKO mice. Interestingly, we found that the mRNA levels of *Ucp1* and *Dio2* were increased in PGAT and SCAT (**Suppl. Figure 1G**). In contrast, other thermogenic markers such as *Pgc1a*, *Prdm16* and *Cidea* were similar or even downregulated in both WAT depots. With advanced age, the mRNA expression of the beigeing markers, *Ucp1* and *Dio2*, became similar between the genotypes while *Prdm16* and *Cidea* (*Pgc1a* in SCAT) were even significantly lower in both WAT depots (**Fig. 2**, see below). This suggests that thermogenesis via beigeing/browning of WAT depots is not increased in AHKO mice and thus does not promote energy expenditure in these mice.

Due to space constrains we have not included the data on aged mice in the manuscript.

Fig 2: Thermogenesis marker in PGAT and SCAT of aged mice. PGAT (left) and SCAT (right) mRNA levels of genes involved in thermogenesis relative to *Cyclophilin* reference gene by qPCR with control group arbitrarily set to 1 for each gene (M, 44wk, HFD, *ad libitum* fed, n=5-6/group). Data represent mean + SEM. Statistical significance was determined by Student's two-tailed *t*-test. $P < 0.05$: * for effect of genotype.

Despite increased *Ucp1* gene expression, we were unable to detect increased UCP1 protein levels in AHKO WAT depots. In fact, we were not able to detect any clear UCP1 protein band in PGAT or SCAT of both genotypes, suggesting that under our tested conditions neither control nor AHKO mice have a detectable UCP1 expression in WAT (Fig. 3 see below; of note the WB is not included in the manuscript). We could only detect unspecific bands close to the predicted size of UCP1. To exclude problems with a specific UCP1 antibody, we also tested a different UCP1 antibody (Cell signaling, 14670; 1:1,000; *data not shown*), but again we only detected unspecific bands. Therefore, we decided not to perform adipose tissue histology since we could not be sure whether a signal would represent UCP1 in our tissue samples.

Overall, our data suggest that adipocytes of AHKO mice do not switch to a brown or beige phenotype in WAT that may contribute to increased energy expenditure. Furthermore, we also measured rectal body temperature in AHKO and control mice. However, body temperature was not different between the genotypes in the *ad libitum* fed state (*data are not included in the manuscript*).

Fig 3: UCP1 protein expression in PGAT and SCAT of middle-aged and aged mice. UCP1 protein expression was detected using Abcam antibody (ab10983; 1:1,000 dilution; 30 μ g protein). Brown adipose tissue was used as positive control (Pos Ctrl BAT; 2 μ g protein).

3. Can the authors suggest a mechanism by which PPAR γ would be simultaneously down-regulated in WAT (Fig. 2) but up-regulated in liver (Fig. 3)?

This is a very interesting question. As shown in several KO-mouse models with defective adipose tissue lipolysis (global HSL-KO mice (5), global ATGL-KO mice (2), adipocyte-specific ATGL-KO mice (6)) the expression of PPAR γ is downregulated in adipose tissue. This indicates that functional adipose tissue lipolysis is essential for PPAR γ mRNA expression.

Similarly, in AHKO mice of this study, middle-aged AHKO mice show a significant downregulation of PPAR γ in both WAT depots (Figure 2H) which goes along with massive reduction in adipose tissue mass (Figure 2F). Thus, the reduced PPAR γ expression reflects the dysfunctional state of the adipose tissue. The upregulation of PPAR γ mRNA expression in the liver of middle-aged AHKO mice (Figure 3F) clearly supports our hypothesis that the uncontrolled FA release via impaired insulin-mediated suppression

of adipocyte lipolysis provokes the fatty liver phenotype in middle-aged AHKO mice (via increased FA uptake and increased expression of genes involved in *de novo* lipogenesis). However, at advanced age the decline in lipolytic activity of the adipose tissue of AHKO mice blunts the uncontrolled FA release and as a consequence also PPAR γ expression in the liver normalizes (**Figure 5A**).

References

- 1) A guide to analysis of mouse energy metabolism. Tschöp MH, et al. Nat Methods. 2011. PMID: 22205519 Free PMC article.
- 2) Hypophagia and metabolic adaptations in mice with defective ATGL-mediated lipolysis cause resistance to HFD-induced obesity. *Schreiber R, et al.* PNAS. 2015, PMID: 26508640.
- 3) Pharmacological inhibition of adipose triglyceride lipase corrects high-fat diet-induced insulin resistance and hepatosteatosis in mice. *Schweiger M, et al.* Nat Commun. 2017 PMID: 28327588
- 4) A recurring problem with the analysis of energy expenditure in genetic models expressing lean and obese phenotypes. *Butler AA, et al.* Diabetes 2010 PMID: 20103710
- 5) Decreased fatty acid esterification compensates for the reduced lipolytic activity in hormone-sensitive lipase-deficient white adipose tissue. *Zimmermann R, et al.* J Lipid Res. 2003.PMID: 12923228.
- 6) Impact of Reduced ATGL-Mediated Adipocyte Lipolysis on Obesity-Associated Insulin Resistance and Inflammation in Male Mice. *Schoiswohl G, et al.* Endocrinology. 2015 PMID: 26196542.

Reviewer #3 (Remarks to the Author):

Pajed et al. provide an complete and complex manuscript. Finding a 'lipodystrophic' phenotype in AHKO mice on a HFD with and observation pointing towards regression of a fatty liver phenotype. The authors provide in depth analyses on the phenotype and provide an interesting conclusion. There are some questions that could be addressen to provide an insight.

Major:

1) The HFD was provided from weaning onwards. While food intake was not different during the 6months of feeding in total kcal's how was the feeding pattern during the day?

We thank the reviewer for this valuable comment. We have now measured food intake of middle-aged and aged mice during light and dark phase (**Suppl. Figure 1D & 3D**). Middle-aged AHKO mice showed no differences in daily food intake. In contrast, aged AHKO mice fed more during the dark phase, whereas total food intake over the whole day was not significantly different.

2) Related to 1), if they eat the same amount but weight less where is the energy going? How about Energy Expenditure? How about fecal loss of energy? Please provide more details on this energy balance.

This is an interesting point. We have now also investigated energy expenditure in middle-aged and aged mice using a laboratory animal monitoring system (PhenoMaster, TSE systems). AHKO mice showed no difference in total energy expenditure (EE) independent of age (**Suppl. Figure 1A & 3A**). Because body weight (BW) is determinant of EE, we performed ANCOVA analyses using BW as a co-variant. When adjusted to a hypothetical BW, the genotypes have also comparable EE (**Fig. 1** see below). Although, we could not detect differences between the genotype, we expect that AHKO mice exhibit a subtle increase in EE and the indirect calorimetric measurement is not sensitive enough to detect these changes (1, 2, 3, 4). The small increase in EE might also explain the longsome development of the lipodystrophic phenotype.

Fig 1. Energy expenditure (EE) in young, middle-aged, and aged mice. A. EE is expressed as adjusted means based on a normalized mouse weight 25.98 g determined using ANCOVA, $P=0.271$ (M, HFD, *ad libitum* fed, $n=5-6$ /group). B. EE is expressed as adjusted means based on a normalized mouse weight 35.41 g determined using ANCOVA, $P=0.654$ (M, HFD, *ad libitum* fed, $n=5-7$ /group). C. EE is expressed as adjusted means based on a normalized mouse weight 43.52 g determined using ANCOVA, $P=0.931$ (M, HFD, *ad libitum* fed, $n=12$ /group). Data represent mean + standard deviation.

To further exclude defective intestinal food absorption as a cause for reduced body weight, we also measured feces mass output and fecal energy content of middle-aged mice via calorimetric analysis

(Suppl. Figure 1E). Both feces mass and fecal energy output were comparable between genotypes, suggesting that impaired intestinal food absorption is not the cause for the observed phenotype in AHKO mice.

3) The liver phenotype of these AHKO mice on a HFD is interesting. Did the authors consider histology to confirm a pathological picture of the increased immune cell and activation?

We have followed up this question and have now included gross images of liver and histological stainings with Oil red O and against F4/80-positive immune cells of middle-aged and aged mice (Figure 4G). The histological images corroborate our observation that with advanced age, the fatty liver phenotype in AHKO mice reverses, including reduced accumulation of neutral lipids and less F4/80-positive cells. We also stained the liver for CAB trichrome (collagenase) but found no differences between genotypes or age (data not shown). Furthermore, we have included plasma ALT measurement into the manuscript (line 238-239), showing that middle-aged AHKO mice tended to have increased ALT levels compared to age-matched control mice ($P=0.074$). With advanced age control mice showed a trend towards increased ALT levels (15.6 ± 2.0 vs 25.5 ± 3.3 , $P=0.057$) compared to middle-aged control mice, while these levels were unchanged in AHKO mice (Fig. 2 see below).

Fig 2: Plasma ALT levels in middle-aged and aged mice (M, HFD, *ad libitum* fed, n=5-9/group). Data represent mean + SEM. Statistical significance was determined by Student's two-tailed *t*-test.

4) The conclusion of remission (abstract) or regression (results/conclusion) seems a little far fetched? If you take a closer look at the TG accumulation you could conclude that there was no difference at 44wks of age, Fig4, D and E. And one other observation is that TG content increases 2-4-fold between 24wks and 44wks. What is the accumulation is progression more rapid in the AHKO mice and reaches a maximum somewhere between 24-44wks? Controls get there later? So no remission or regression but simple reaching maximal capacity without further increase possibility? A histological picture of the livers would provide a better insight as well.

This is an interesting point. We have performed additional experiments using two new cohorts (middle-aged and aged mice) to address this. The data show that with advanced age of control mice (middle-aged vs aged mice) liver steatosis develops (increased liver weight, TG content; Figure 4E, 4G). In contrast, AHKO mice at this period of age develop progressive lipodystrophy (Figure 4B), so that their already existing fatty liver phenotype does not further exacerbate. A direct comparison of liver weight (Figure 4D), gross appearance, histological ORO staining of liver sections (Figure 4G), and hepatic TG content (Figure 4E) between middle-aged and aged mice support the notion that the fatty liver phenotype improves. This is supported by reduced liver weight, apparently less ORO staining of liver section and tentatively less hepatic TG content of aged as compared to middle-aged AHKO mice. This is also evident when the fatty liver phenotype of aged AHKO mice is compared to age matched control mice.

5) The conclusion of the data in fig6 might require a different one too? Looking at the glucose graphs during the GGT it looks like there is better clearance but raises a couple of questions: a) the dose was per kg body weight and the AHKO mice are lighter compared to the controls. Did the authors do

a GGT with a equal dose of glucose? b) secondly, based on the fed data the insulin response after a challenge seems higher, so please provide the insulin data during the GTT's. If there is more insulin during a GTT and a real increased glucose clearance this indicates either insulin resistance or insulin sensitivity.

We have performed GTTs using 1.6 g glucose/kg body weight (Figure 6D, 6E; the glucose dose was calculated for each mouse individually). However, we have also determined glucose clearance when given the same glucose dose to all experimental animals. These data are not included in the manuscript. We found that AHKO mice are also under these conditions more glucose tolerant (Fig. 3 see below). We further determined plasma insulin levels before (6h fasted; time point 0, Figure 6C) and following a glucose bolus (at time point 15 min of the GTT, Figure 6C). Both middle-aged and aged AHKO mice showed reduced plasma insulin levels under basal (time point 0) and glucose stimulated conditions (time point 15 min). Overall, these data indicate that both middle-aged and aged AHKO mice have increased glucose clearance, but that with advanced age and in the presence of an “improved” liver phenotype compared to age-matched control mice, aged AHKO mice exhibited a better glucose clearance.

Fig 3: GTT in middle-aged (left) and aged (right) mice (using the same glucose dose [75 mg glucose per mouse]) (M, HFD, 6h-fasted, n=4-9/group. Data represent mean \pm SEM. Statistical significance was determined by Student's two-tailed t-test. $p < 0.05$; * for effect of genotype.

Of note, the reduced plasma insulin levels in the fasted state (12h-fast in Figure 6B and 6h-fast in Figure 6C, respectively) might be caused by the reduced levels of fasted FAs in the plasma of these mice. Previous studies showed that plasma FAs are essential for insulin secretion in the fasted state (5).

6) Related to 5), the pAKT data is not provided proof for more insulin sensitivity, fig6. How do the basal pAKT levels look like? Please provide the increase compared to unstimulated tissue. For true IR or IS measurements consider the euglycemic hyperinsulinemic clamps.

As suggested by the reviewer, we have now added for all insulin signaling studies Western blots for basal AKT phosphorylation. We have included these data in the quantification of the pAKT (AKT^{pSer473}) in WAT (Figure 3D; Suppl. Figure 3E) and liver (Suppl. Figure 4E).

Furthermore, we agree that for more detailed insulin resistance/sensitivity measurements, euglycemic hyperinsulinemic clamp measurements are presumably the gold standard. Such measurements are in particularly important in the AHKO mouse model since these mice become more glucose tolerant over age at a time point when lipodystrophy progresses. Unfortunately, we do not have the opportunity to perform such clamp studies in our facility.

Minor

1) Some of the data is provided from female mice and other from male mice? Is there any indication that there is no sex-difference?

Initially, we have also included female mice. We observed a comparable lipid phenotype to that of male mice. For example, we found similarly reduced body weight and fat mass when fed HFD. These changes in female mice, however, started at a somewhat later time-point. Furthermore, both female and male AHKO mice showed a significant reduction in basal and stimulated glycerol release. We have now eliminated all data from female mice and replaced them with data from male mice. This should make the data even more consistent.

References

- 1) Hypophagia and metabolic adaptations in mice with defective ATGL-mediated lipolysis cause resistance to HFD-induced obesity. *Schreiber R, et al.* PNAS. 2015, PMID: 26508640.
- 2) Pharmacological inhibition of adipose triglyceride lipase corrects high-fat diet-induced insulin resistance and hepatosteatosis in mice. *Schweiger M, et al.* Nat Commun. 2017 PMID: 28327588
- 3) A recurring problem with the analysis of energy expenditure in genetic models expressing lean and obese phenotypes. *Butler AA, et al.* Diabetes 2010 PMID: 20103710
- 4) A guide to analysis of mouse energy metabolism. *Tschöp MH, et al.* Nat Methods. 2011. PMID: 22205519 Free PMC article.
- 5) Essentiality of circulating fatty acids for glucose-stimulated insulin secretion in the fasted rat. *Stein DT, et al.* J Clin Invest. 1996. PMID: 8675683

REVIEWERS' COMMENTS:

Reviewer #1 (Remarks to the Author):

I have no further comments. The authors have done an exceptional job at addressing all of the concerns.

Reviewer #2 (Remarks to the Author):

The authors have responded to the concerns with additional experimental data, which is much appreciated. That's the good news. The bad news is that the new data puts us no closer to understanding the mechanism underlying this phenotype than we were before. The mice have reduced adiposity, but this is not explained by reduced food intake, elevated energy expenditure, or reduced lipid absorption. The mice seem to violate the laws of thermodynamics! The authors believe that there is a small increase in energy expenditure that they can't measure, and while that's technically possible, it seems unlikely for an effect of this magnitude.

One relatively simple thing they should assess is whether stromal cells isolated from the fat pads of these mice are able to differentiate normally *ex vivo* (they need to confirm that HSL is knocked out in this model, even though the Cre may work late in the process). This can be supported by simply knocking down HSL in 3T3-L1 preadipocytes and assessing differentiation. If there's a defect in adipogenesis then that would account for the lipodystrophy seen in young and middle-aged animals. It wouldn't clarify why things get better as they age, though.

I think the manuscript describes a very interesting phenotype. I don't think I understand how it comes to pass. Whether this is sufficient for Communications Biology is an editorial decision.

Reviewer #3 (Remarks to the Author):

I thank the Authors for there thorough rebuttal with additional experiments.

The only question which can be added for discussion would be: How does this phenotype hold with either thermo-neutrality or cold exposure. This might give further insight where the calories are going, thermogenesis?

Warm regards

Ms number/ID: COMMSBIO-20-0840-B

We thank the reviewers for their constructive comments.

REVIEWERS' COMMENTS:

Reviewer #1 (Remarks to the Author):

I have no further comments. The authors have done an exceptional job at addressing all of the concerns.

Thank you.

Reviewer #2 (Remarks to the Author):

The authors have responded to the concerns with additional experimental data, which is much appreciated. That's the good news. The bad news is that the new data puts us no closer to understanding the mechanism underlying this phenotype than we were before. The mice have reduced adiposity, but this is not explained by reduced food intake, elevated energy expenditure, or reduced lipid absorption. The mice seem to violate the laws of thermodynamics! The authors believe that there is a small increase in energy expenditure that they can't measure, and while that's technically possible, it seems unlikely for an effect of this magnitude.

One relatively simple thing they should assess is whether stromal cells isolated from the fat pads of these mice are able to differentiate normally ex vivo (they need to confirm that HSL is knocked out in this model, even though the Cre may work late in the process). This can be supported by simply knocking down HSL in 3T3-L1 preadipocytes and assessing differentiation. If there's a defect in adipogenesis then that would account for the lipodystrophy seen in young and middle-aged animals. It wouldn't clarify why things get better as they age, though.

I think the manuscript describes a very interesting phenotype. I don't think I understand how it comes to pass. Whether this is sufficient for Communications Biology is an editorial decision.

We thank the reviewer for the comments.

Concerning differentiation of stromal cells: To evaluate whether stromal cells from AHKO mice differentiate normally, we isolated stromal vascular fraction from subcutaneous adipose tissue derived from control and AHKO mice and differentiated cells to primary adipocytes (see **Figure 1** below). In control cells, mRNA expression of *Hsl* increased following the differentiation process (from day 0 to day 10). In AHKO cells, *Hsl* mRNA levels were similar at day 0 and day 2 but started to reduce at day 4 compared to control cells. On day 10 of differentiation (primary adipocytes) *Hsl* expression was -90 % reduced compared to control primary adipocytes (**Figure 1a**). This expression pattern was expected since deletion of HSL is regulated by Cre recombinase under the control of Adiponectin promoter and the expression of Adiponectin started to increase at day 4 of differentiation (**Figure 1b**). Remarkably, stromal cells derived from AHKO mice were capable to differentiate into primary adipocytes in a cell autonomous manner (*see experimental condition in the Supplementary Method-Section*). Primary

Adipocytes from control and AHKO mice showed comparable mRNA levels of target genes involved in adipogenesis (**Figure 1c**). Furthermore, AHKO primary adipocytes exhibited normal lipid accumulation (**Figure 1d**). Thus, adipocyte differentiation ability is not impaired in AHKO cells *per se*, which most likely results from the “late” deletion of HSL in primary cells. However, since WAT of AHKO mice show reduced mRNA expression of target genes involved in lipo/adipogenesis in middle-aged (*Figure 2h in the manuscript*) and aged mice (data not shown), we conclude that with progression of the lipodystrophic phenotype the ability to differentiate new adipocytes within the adipose tissue diminishes probably due to reduced availability of signaling lipids for the differentiation process. Similar results were found in ATGL-KO mice (1).

Due to limited space, we did not include these data into the manuscript.

Fig. 1: Differentiation of stromal vascular fraction to primary adipocytes. a-b. Relative mRNA expression of *Hsl* (a) and *Adiponectin* (b) during differentiation. c. Relative mRNA expression of *Pparg2* (left) and *Cebpa* (right) in preadipocytes (day 0) and adipocytes (day 10). d. Left: Oil red O staining of preadipocytes and adipocytes. Right: Photometric quantification of the extracted ORO dye in preadipocytes and adipocytes. Data represent mean + SEM. Statistical significance was determined by Student’s two-tailed *t*-test. $P < 0.05$: * for effect of genotype.

Concerning reduced adiposity and energy expenditure: Although we did not observe that the energy expenditure was increased in our mouse model, we hypothesize that AHKO mice have increased energy expenditure, but that the differences are too small to measure (e.g. We see a trend towards increased locomotor activity in aged mice, which might pin-point towards increased energy expenditure. Similarly, also adipose tissue loss of aged mice may promote heat loss). In line with this assumption, a previous study with global HSL-KO mice fed HFD mice showed that these mice have increased energy expenditure (2). Overall, we agree with reviewer’s opinion that further studies are necessary to investigate the fate of energy in AHKO mice.

- 1) Hypophagia and metabolic adaptations in mice with defective ATGL-mediated lipolysis cause resistance to HFD-induced obesity. *Schreiber R, et al. PNAS. 2015, PMID: 26508640*
- 2) Attainment of brown adipocyte features in white adipocytes of hormone-sensitive lipase null mice. *Ström K. et al. PLoS One PMID: 18335062*

Reviewer #3 (Remarks to the Author):

I thank the Authors for there thorough rebuttal with additional experiments.

The only question which can be added for discussion would be: How does this phenotype hold with

either thermo-neutrality or cold exposure. This might give further insight where the calories are going, thermogenesis?

We thank the reviewer for this comment.

We agree that it is important to unveil the fate of energy in AHKO mice. Therefore, we have already started a new project where we investigate the impact of thermoneutrality and cold exposure on the energy metabolism in these mice. So far, we confirmed previous data from global HSL-ko mice (1) showing that AHKO mice (*on standard chow diet*) are not cold-sensitive whether in the presence (*data not shown*) or absence of food (**Figure 2a**). Due to reduced adipose tissue lipolytic activity in AHKO mice, body weight loss was impaired upon fasting at 4°C (**Figure 2b**). Blood glucose levels were similar in control and AHKO mice both before and after cold exposure (**Figure 2c**). Fatty acid levels were comparable in the fed state before cold exposure but significantly reduced upon fasting in the cold, whereas triglyceride levels did not differ between the genotypes (**Figure 2d**). Future studies with AHKO mice fed HFD housed at 4°C or at thermoneutrality will provide additional information on the impact of energy expenditure to the observed phenotype of AHKO mice. However, these studies are beyond the scope of the revision process.

Fig. 2: Cold exposure of AHKO mice. **a.** Rectal body temperature of control and AHKO mice in the fasted state upon cold exposure. **b.** Body weight loss after 6 h fasting in the cold. **c.** Blood glucose and **d.** plasma lipid parameter before and 6 h fasting in the cold (M, chow diet, 16wk, n=6/group). Data represent mean + SEM. Statistical significance was determined by Student's two-tailed *t*-test. *P* < 0.05: * for effect of genotype; # for effect of treatment.

- 1) Targeted disruption of hormone-sensitive lipase results in male sterility and adipocyte hypertrophy, but not in obesity. *Osuga J, et al. PNAS. 2000, PMID: 106391588*